# CGBENCH: Benchmarking Language Model Scientific Reasoning for Clinical Genetics Research

**Owen Queen**
Stanford University
oqueen@stanford.edu

**Harrison G. Zhang**
Stanford University
hgzhang@stanford.edu

**James Zou**
Stanford University
jamesz@stanford.edu

## Abstract

Variant and gene interpretation are fundamental to personalized medicine and translational biomedicine. However, traditional approaches are manual and labor-intensive. Generative language models (LMs) can facilitate this process, accelerating the translation of fundamental research into clinically-actionable insights. While existing benchmarks have attempted to quantify the capabilities of LMs for interpreting scientific data, these studies focus on narrow tasks that do not translate to real-world research. To meet these challenges, we introduce CGBENCH, a robust benchmark that tests reasoning capabilities of LMs on scientific publications. CGBENCH is built from ClinGen, a resource of expert-curated literature interpretations in clinical genetics. CGBENCH measures the ability to 1) extract relevant experimental results following precise protocols and guidelines, 2) judge the strength of evidence, and 3) categorize and describe the relevant outcome of experiments. We test 8 different LMs and find that while models show promise, substantial gaps exist in literature interpretation, especially on fine-grained instructions. Reasoning models excel in fine-grained tasks but non-reasoning models are better at high-level interpretations. Finally, we measure LM explanations against human explanations with an LM judge approach, revealing that models often hallucinate or misinterpret results even when correctly classifying evidence. CGBENCH reveals strengths and weaknesses of LMs for precise interpretation of scientific publications, opening avenues for future research in AI for clinical genetics and science more broadly.

○ Code   🤗 Benchmark Data

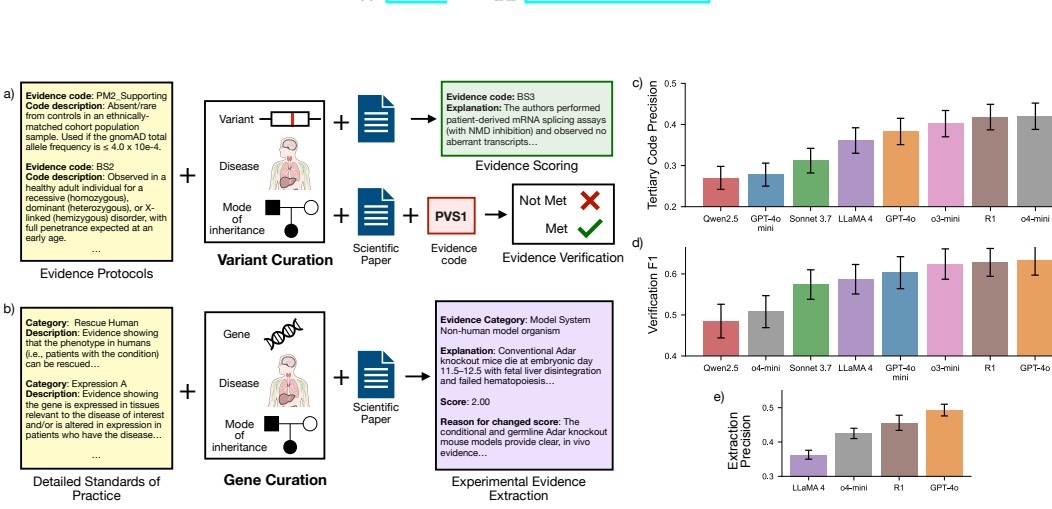

Figure 1: Overview of CGBENCH tasks and results. a) Variant Curation benchmarking tasks, which encompass evidence scoring (E-Score) and evidence verification (E-Ver). b) Gene Curation benchmarking workflow for annotating and scoring experimental evidence. c) E-Score results for tertiary codes: reasoning models show the highest performance. d) E-Ver results: GPT-4o scores highest and more mixed results for reasoning models. e) Experimental evidence extraction results.

39th Conference on Neural Information Processing Systems (NeurIPS 2025) Track on Datasets and Benchmarks.

# 1 Introduction

Advancements in sequencing technologies have brought about a revolution in clinical genetics and personalized medicine [1]. In the field of clinical genetics, genetic testing is integrated with expert interpretation in order to aid in the diagnosis, management, and counseling of patients with conditions that have genetic causes. As such, we are able to understand how variations in the human genome may contribute to the risk of certain disease states. A key challenge in translating genomic data into clinical insights is determining 1) which genes are causal for a disease (*gene curation*) and 2) which genetic variants within a gene are associated with diseases, i.e., are pathogenic for a disease (*variant curation*). These relationships can be elucidated through many approaches, which can include laboratory experiments in disease models, genome-wide association studies in large patient populations [2, 3], and analysis of population frequencies [4, 5]. Traditional approaches in establishing gene-disease validity or variant pathogenicity for a disease is a labor-intensive task that requires manual review of mountains of evidence across multiple studies. Evidence from different studies might be contradictory or inconclusive, raising the possibility of misinterpretation and misguided clinical practice. Thorough and precise interpretation of diverse evidence is crucial to translation of basic genetic and biological research into actionable clinical insights.

To address these challenges, the Clinical Genomic Resource (ClinGen) is a National Institutes of Health (NIH)-funded resource that evaluates and curates the clinical relevance of genetic variants [6, 7]. ClinGen convenes large groups of expert curators highly skilled in biomedical evidence synthesis and literature analysis to collect reputable evidence used to determine if sufficient evidence is available to establish pathogenic of a variant [8] or validity of a gene-disease association [9]. ClinGen oversees the establishment of U.S. Food and Drug Administration (FDA)-approved rigorous guidelines and procedures for gene and variant curation.

Generative large language models (LMs) present an exciting opportunity for advancing evidence curation in clinical genetics. LMs excel in interpretation of scientific publications [10, 11], and facilitating multi-step complex research tasks [12, 13, 14]. High-throughput literature interpretation and synthesis can be a valuable tool in curating bodies of evidence for annotating variants and genes. As LMs are increasingly deployed in scientific research pipelines, there is a need for datasets and evaluation procedures that realistically capture the complex nature of practical scientific research.

Many benchmarks have been established to evaluate the capabilities of LMs to perform scientific tasks and to interpret scientific literature, especially for biomedical research [15, 16, 17, 10, 18, 19, 20, 21]. However, many of these works have focused on contrived or narrow tasks within the scientific domain, and very few are reflective of real-world, translational scientific uses. Additionally, many benchmarks focus on simple queries such as multiple choice questions or simple assertion verification, tasks that do not capture the breadth of capabilities of LMs. Thus, a gap exists in datasets and tools to evaluate the ability of LMs and agents to perform complex evidence synthesis tasks on an expert-level.

**Our contributions.** We present CGBENCH, a large-scale benchmark for evaluation of LM capabilities in discerning and classifying scientific evidence in clinical genetics. CGBENCH is sourced from ClinGen's Evidence Repository (ERepo) [6, 22, 7, 23], which contains thousands of curations of gene and variant annotations. We aggregated annotations across the ERepo and provide succinct tasks to measure the capabilities of LMs for assisting in variant and gene curation processes. CGBENCH presents tasks and methods to measure the ability of LMs to extract, discern, and explain findings within scientific papers that are relevant and match given protocols that are specific to disease, gene, and/or variant context. CGBENCH utilizes detailed and thorough guidelines developed for review of genes and variants to prompt LMs for these complex tasks.

We benchmarked 8 different LLMs and a variety of prompting techniques to understand the capabilities of LLMs for this task. We uncovered some distinct results: 1) many LLMs zero-shot are relatively weak at these complex tasks, 2) prompt optimization can significantly increase the performance of LLMs, 3) reasoning models, such as Deepseek-R1 and o4-mini, provide a notable boost over non-reasoning models in extracting relevant evidence but suffer comparatively in discerning weak from strong evidence. We use an LM-as-a-judge approach to evaluate concordance of evidence interpretations. Our results show that even when controlling for matched explanations, LMs sometimes still fail to match ClinGen explanations. We release CGBENCH (code[1] and data[2])

---

[1]https://github.com/owencqueen/cgbench
[2]https://huggingface.co/datasets/owencqueen/cgbench_data/

as an open-source resource for the community to explore rigorous evaluation of LMs and agents on literature interpretation for translational clinical genetics research.

## 2 Related work

**Scientific understanding benchmarks.** Traditional benchmarks for measuring LLM literature synthesis have focused on claim verification within papers [15, 16, 17] or extractive multiple-choice questions [10, 18]. Others focus on general scientific tasks [19, 20, 21], multimodal comprehension of figures and tables [24, 25], or generation of peer reviews [26, 27]. However, few datasets have access to ground-truth explanations, although recent efforts have focused on including expert explanations [28]. Most importantly, many datasets do not focus on tasks that accurately reflect real-world use cases for LMs in scientific research. We set forth to tackle these challenges in CGBENCH.

**Agents and LMs for biomedical discovery.** LMs are increasingly being integrated into scientific discovery pipelines [29, 30]. Many biomedical LMs have been developed [31, 32, 33], but recent studies have shown that domain-specific LMs often do not outperform specially-prompted general LMs [34] outside of a few proprietary models [35, 36]. General search agents have been developed to comb large corpora of scientific papers and provide literature searches [37, 14, 10, 38]. Multi-agent systems have been built that can conduct complex scientific research, utilizing tools and multi-agent interaction [39, 40, 41]. Recent efforts have demonstrated the use of AI agents in designing antibiotics [42], validating AI research hypotheses [43], and designing novel SARS-CoV2 nanobodies [12].

**Variant and gene curation with LMs and agents.** Significant past efforts have focused on mining of literature that references variants or genes. This has been done through traditional literature mining [44, 45, 46], deep-learning based retrieval [47], and more recently, agentic LM approaches [48, 49]. These works all contain some prioritization method for papers, but these are often based on loose criteria such as filters to ensure a gene is a central experimental focus rather than mentioned in related work. LitGen [50] is the closest to our setup, but it fails to account for complexities of VCEP specifications (Section 3). Other related works are discussed in Appendix C.

## 3 CGBENCH Benchmark

CGBENCH consists of expert-annotated variant and gene curations that are sourced from the ClinGen Evidence Repository (ERepo) at `clinicalgenome.org`. The data from the ERepo consists entirely of human curator-reviewed entries, ensuring that data are of the highest quality. CGBENCH pools elements from across the ERepo, VCI criteria specification registry, and GCI standards of practice and aggregates them into single inputs for LM prompts (discussed concretely in Section 3.3). In this section, we discuss the core concepts behind CGBENCH and lay out the components relevant to the benchmarking tasks. We then describe the structure and formulation of benchmarking tasks.

### 3.1 Variant Curation Interface (VCI)

The ClinGen VCI is an open-source, expert-curated variant classification platform that supports the application of evidence criteria and classification of variants based on variant classification guidelines developed by the American College of Medical Genetics and Genomics (ACMG)/Association for Molecular Pathology (AMP) [22, 8] (see Appendix C for more information). The procedure for the VCI, as originally defined in [8], consists of gathering evidence (primarily papers) and assigning "evidence codes" that define 1) the type of evidence, such as whether it comes from an association study or from experimental data, and 2) the strength of the evidence. CGBENCH focuses on evaluating the evidence scoring of scientific literature, as given by PubMed IDs in the VCI database. More information of our curation of the VCI can be found in Appendix D.

**Assigning evidence codes.** The assignment of evidence codes is a complex task that requires deep understanding of the biological and clinical implications of a scientific paper. First, evidence codes are structured in a hierarchy [8], as shown in Figure 2. The primary code first defines if the evidence leans towards pathogenicity or benignity. The secondary code then describes relative strength of the code. Finally, the tertiary code describes the type of evidence presented. Assigning an evidence code requires not only fine-grained interpretation of evidence but also determining the evidence's strength. An example of a code is "PP4", a pathogenic (P) "supporting" (P) strength code that is centered around determining if a "Patient's phenotype or family history is highly specific for a disease with a

single genetic etiology", or "BS3", a benign (B) "strong" (S) code that is satisfied if "Well-established in vitro or in vivo functional studies show no damaging effect on protein function or splicing."

Code specifications change depending on the variant and disease being examined. Variant curation expert panels (VCEPs) establish the guidelines that are regularly updated and expanded. While many codes are shared across VCEPs, this requires LMs to adapt to different guidelines across variants and diseases, requiring strong generalization to new instructions. Appendix D contains more details about VCEPs.

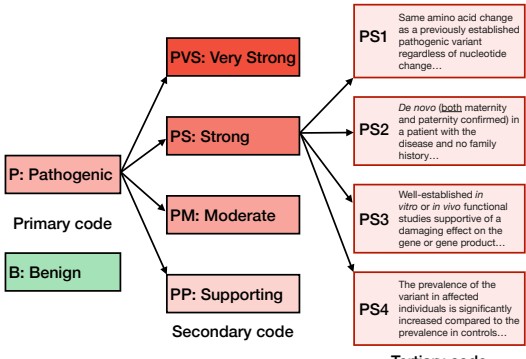

Figure 2: VCI evidence code hierarchy.

### 3.2 Gene Curation Interface (GCI)

The GCI [23] gene-disease validity annotations, with a comprehensive set of protocols for determining whether sufficient evidence is met to establish clinical validity of a gene and disease relationship [9]. Similar to the VCI, the GCI curation procedure is built on gathering and assessing the strength of evidence.

**Experimental evidence extraction.** We focus on extracting "Experimental Evidence" entries, which are defined as "experimental information necessary to determine whether the function of the gene product is at least consistent with the disease with which it is associated, if not causally implicated" [9]. The experimental evidence is entered into the GCI as a table with four elements. 1) **Evidence category**: the classification of evidence into 13 different distinct classes. 2) **Explanation**: a description of the evidence described in the paper. 3) **Score**: a floating point value assigned based on the strength of the evidence presented. The SOP provides a default score and range of acceptable scores for each evidence category. 4) **Reason for changed score**: If the score is not given as the default score, this field explains why the evidence was weaker or stronger than expected.

The protocols defining these categories and how to assign their strength are defined in standards of practice (SOPs) published by ClinGen. Like the VCI, interpreting these protocols requires the LM to deeply understand the experimental protocols and findings of the results, as the strength component may be defined by factors such as sample size or similarity of the model to human biology.

### 3.3 CGBENCH tasks and formulations

CGBENCH consists of three primary tasks: evidence scoring, evidence verification, and evidence extraction. We now describe how these tasks can be formulated succinctly to evaluate LM capabilities for scientific interpretation.

**VCI definitions.** We first define a few shared concepts for the VCI tasks. Formally, denote the variant annotation query $q_i^v$ as a tuple of input variables $q_i^v = (d_i, v_i, m_i)$ where $d_i$ is the disease, $v_i$ is the variant, and $m_i$ is the mode of inheritance for the variant. Note that $v_i$ inherently encodes the gene, as we use the HGVS nomenclature provided by ClinGen Evidence Repository [51, 22]. Then represent the text as $T_j$; a different index $j$ is used to denote that one text may apply to multiple $q_i^v$ queries. Finally, an evidence code $y_k$ is drawn from a discrete set $\mathcal{Y}_{\text{vcep}}$ and is accompanied by a description $e_k^y$ also defined by the VCEP used to categorize the variant. For all definitions, we will assume "vcep" refers to the relevant VCEP for the gene and disease.

**Task 1: VCI Evidence Scoring (E-Score).** This task is concerned with extracting meaningful conclusions from a scientific paper based on detailed, precise instructions specific to the gene and disease being examined, e.g., the VCEP specifications (Figure 1a). The task can be written as a function **ES**:

$$\mathbf{ES}\left(q_i^v, T_j, \mathcal{Y}_{\text{vcep}} | f_{\text{LM}}\right) = \hat{y}_k; \hat{y}_k \in \mathcal{Y}_{\text{vcep}} \tag{1}$$

Where $\hat{y}_k$ is the predicted evidence code. We additionally prompt the LLM for an explanation for its classification, but for the multiclass classification problem, we can score the ability to choose the correct $y_k$ from $\mathcal{Y}_{\text{vcep}}$.

VCI Escore covers 239 total samples, of which we use 205 for evaluation. This task covers 120 unique PubMed papers, 40 diseases, and 191 variants. There are 33 total VCEPs included in this task, averaging only 7.24 samples per VCEP; across these samples, 13 unique evidence codes are covered.

**Task 2: VCI Evidence Verification (E-Ver).** The model is asked to determine, given an evidence code, if a code is "met" or "not met" from the publication (Figure 1a). This is a function **EV**:

$$\mathbf{EV}\left(q_i^v, T_j, y_k | f_{\mathrm{LM}}\right) = \hat{v}; \hat{v} \in \{\text{"met"}, \text{"not met"}\}, y_k \in \mathcal{Y}_{\mathrm{vcep}} \tag{2}$$

There are 286 total examples for the E-Ver task, with 242 being used for evaluation; of those 167 codes are "not met" and 119 are "met". This covers 191 variants and 39 diseases, 39 unique VCEPs and 28 evidence codes, over 2x the number of codes covered in E-Score.

**Task 3: GCI Experimental Evidence Extraction.** For the GCI, we're interested in extracting concise pieces of evidence from publications that are relevant to a gene-disease validity determination, similar to the tables provided in the GCI (Figure 1b). For extracting experimental evidence in the GCI, we first define a gene-centric query $q_i^g$ as a tuple of $(d_i, g_i, m_i)$, where $d_i$ is a disease, $g_i$ is a gene, and $m_i$ is a mode of inheritance. $T_j$ is a reference PubMed article. The output is structured set of properties (defined in Section 3.2): $a_i$ is an evidence category, $h_i$ is an explanation of the evidence extracted, $s_i$ is the score, and $r_i$ is a reasoning for why the score was changed from default for this SOP. Note that $r_i$ might be empty if $s_i$ is equal to the default score. The criteria specification is given by $\mathcal{C}_{\mathrm{sop}}$, as defined by the SOP under which the entry was originally annotated. The set of extracted evidence units for example $i$ are tuples $(a_i, h_i, s_i, r_i) \in \mathcal{E}_i$; the output is a set of extracted evidence units $\hat{\mathcal{E}}_i$. Thus, the task is denoted by a function **EE**:

$$\mathbf{EE}\left(q_i^g, T_j, \mathcal{C}_{\mathrm{sop}} | f_{\mathrm{LM}}\right) = (a_i, h_i, s_i, r_i) \tag{3}$$

We then can measure the correspondence of each element in the output, as is described in Section 4.2 and Section 4.3. The GCI experimental evidence includes many more examples than the VCI tasks at 2155 total. We split this into 336 samples by taking a date-based split: for all entries entered after July 7, 2024, we include them in the evaluation set, all others are used for potential in-context examples (size of 1819 samples). These annotations in total cover 860 diseases and 1291 genes. There are 13 unique experimental categories across these samples, which are described in Appendix D.

# 4 Benchmarking Methods

We now describe how we formulate and prompt LMs to perform CGBENCH tasks. Finally, we discuss a few advanced prompting techniques and an LM-as-a-judge approach to evaluate explanations.

## 4.1 Prompting LMs for CGBENCH tasks

We construct prompts for LMs based on the following approaches: 1) "chain-of-thought" prompting [52], 2) role-playing prompts to inform the model that it is skilled in clinical genetics, and 3) full context described in Section 3.3 to ensure the models have access to the gene/variant information and full-text of the paper it is asked to examine. Genes are given as HGNC names [53] and variants and given in HGVS nomenclature [51]. All prompts also request a precise structure that are described throughout Appendix K. Additionally, we employ pass@5 sampling for the VCI E-score task and the GCI evidence extraction task. Find full details in Appendix K.

**In-context learning for literature interpretation.** In-context learning describes ability of LMs to learn complex functions by simply observing demonstrations of the task in the prompt of the LM [54]. Given the complexity of the tasks presented in CGBENCH, we explore the use of in-context learning to improve LM performance. Each base prompt (i.e., what would constitute a demonstration) consists of a full-length text of a paper, and this can quickly balloon the size of the context for the model in the in-context regime. Thus, we use demonstrations that omit the full-text of the paper, showing only the evidence codes or extracted evidence and explanations given in ClinGen. Our hypothesis is that by showing the LM examples of why different evidence codes would be met, the LM will understand scenarios in which a code may apply. This allows us to expand up to 30 examples for E-Score. Given the complexity of the GCI evidence extraction task, we use 13 in-context examples in all prompts that include at least one of every instance of an extracted evidence category.

## 4.2 Evaluating LM-extracted evidence

**VCI.** *Evidence-Scoring* is a multilabel classification problem, so we score the ability to retrieve against potentially-multiple codes assigned to an input query and paper. Our metrics are ***precision*** and ***recall***, measured over the evidence codes assigned to a given $T_j$ for a query $q_i^v$. Precision is intuitively defined by the question "of the codes predicted, how many are in the ground-truth?", and recall is "of the codes in the ground-truth, how many of them did I identify?". *Evidence Verification* is a binary classification problem, so we report true positive rate, true negative rate, and F1.

**GCI Evidence Extraction Scoring.** For the experimental evidence extraction, we use five metrics to measure three capabilities of the LM for this task. **1) Category Matching**: In this case we ask: can the LM identify the correct categories of evidence from the paper? We use precision and recall as metrics as this is a multilabel classification problem. **2) Structured output adherence**: Adherence to structured output is critical for parsing the multiple tuples needed for evidence extraction. Thus, we measure adherence to desired structure via measuring success rate of our parsing function (Appendix L). **3) Normalized MAE**: We next evaluate LM abilities to score evidence according to guidelines given by ClinGen's GCI. Some experimental categories, such as "Rescue in Human" (range 0-4), have larger ranges than others, such as "Cell culture model" (range 0-2). Normalized MAE measures the mean absolute error (MAE) normalized to the range of the scores. **4) $\Delta$Strength**: This metric focuses only on the sign change with respect to the default score, e.g., if the score is decreased, increased, or held equal with respect to the default score. This measures if the LM correctly up- or down-weighted the strength of a piece of evidence relative to what was interpreted by GCI curators. The metric is proportion of samples that match the expected sign change. Note that we compute Normalized MAE and $\Delta$Strength only on category-matched samples to ensure evaluation only across likely matches.

## 4.3 Evaluating LM explanations against ground-truth explanations

CGBENCH includes 2680 explanations given by expert curators; we use these to evaluating the agreement between the ERepo and LMs. We employ a robust LM-as-a-judge approach using best practices from recent literature and use this to measure the correspondence of LM explanations vs. those given by experts in the curation process available through ClinGen.

**Purpose of explanation evaluation.** We are roughly interested in the question of "Do explanation $e_{LM}$ and explanation $e_{curator}$ refer to the same piece of evidence?", where $e_{LM}$ is an explanation generated by an LM and $e_{gt}$ is an explanation given by ClinGen for why either an evidence code is met/not met (VCI) or explaining an extracted piece of evidence (GCI). Thus, our primary interest is in determining if two explanations 1) refer to the same piece of evidence and 2) make the same conclusions about the evidence. In this, case, we are not concerned with preferring any explanation over another as is a popular setup [55]. We note that this method does not account for valid alternative interpretations of the evidence provided by the LM; thus, we view this approach as measuring correspondence to human interpretation of evidence rather than absolute "correctness".

**LM-as-a-judge prompting.** We utilize an LM-as-a-judge approach to automatically evaluate LM explanations against human explanations [56]. Our prompts borrow from standard practices in LM judges, such as avoiding positional biases [57] and chain-of-thought prompting [58] (see Appendix I).

Specifically, we test out three different methods of prompting. First, **task-agnostic** prompting uses a general prompt that does not include any context about the task, only the two explanations. Second, **task-aware** prompting gives the relevant query information to the judge but not the PubMed paper associated with the input. Finally, **evidence-aware** prompting gives the relevant query as well as the PubMed paper for the input to the judge; this essentially gives the judge full awareness of the evidence from which the explanations were pulled. We test all of these methods to understand if the addition of task-specific and evidence information increases or decreases bias with respect to data gathered in the user study. Specific details on judge prompts are elaborated on in Appendix I.

**Calibration with manual review.** To calibrate the LM judge approach, we perform a manual review of a subset of LM and ground-truth explanations; review was assisted by a medical trainee with experience in clinical genetics research. We use this to select the judge approach among the above described methods based on correspondence to the manually-reviewed set.

# 5 Results

| Method | Evidence Scoring | | | | | |
| | Primary | | Secondary | | Tertiary | |
| | Precision@5 | Recall@5 | Precision@5 | Recall@5 | Precision@5 | Recall@5 |
|---|---|---|---|---|---|---|
| Random | $0.524_{\pm 0.219}$ | $0.980_{\pm 0.140}$ | $0.144_{\pm 0.155}$ | $0.550_{\pm 0.497}$ | $0.038_{\pm 0.083}$ | $0.180_{\pm 0.384}$ |
| GPT-4o-mini | $\underline{0.841}_{\pm 0.025}$ | $0.849_{\pm 0.025}$ | $0.463_{\pm 0.032}$ | $0.527_{\pm 0.034}$ | $0.278_{\pm 0.028}$ | $0.341_{\pm 0.032}$ |
| GPT-4o | $\mathbf{0.861}_{\pm 0.024}$ | $\underline{0.878}_{\pm 0.023}$ | $\mathbf{0.517}_{\pm 0.033}$ | $0.568_{\pm 0.033}$ | $0.383_{\pm 0.032}$ | $0.427_{\pm 0.033}$ |
| Sonnet 3.7 | $0.828_{\pm 0.025}$ | $0.868_{\pm 0.024}$ | $0.428_{\pm 0.032}$ | $0.483_{\pm 0.034}$ | $0.312_{\pm 0.030}$ | $0.359_{\pm 0.032}$ |
| Qwen2.5 72B | $0.807_{\pm 0.024}$ | $0.863_{\pm 0.024}$ | $0.481_{\pm 0.031}$ | $0.559_{\pm 0.034}$ | $0.270_{\pm 0.028}$ | $0.322_{\pm 0.032}$ |
| LLaMA 4 | $0.837_{\pm 0.023}$ | $0.873_{\pm 0.023}$ | $0.471_{\pm 0.032}$ | $0.532_{\pm 0.034}$ | $0.361_{\pm 0.031}$ | $0.424_{\pm 0.034}$ |
| Deepseek R1 | $0.780_{\pm 0.023}$ | $\mathbf{0.898}_{\pm 0.021}$ | $0.485_{\pm 0.030}$ | $\mathbf{0.629}_{\pm 0.033}$ | $\underline{0.418}_{\pm 0.031}$ | $\mathbf{0.517}_{\pm 0.034}$ |
| o3-mini | $0.768_{\pm 0.027}$ | $0.844_{\pm 0.025}$ | $0.482_{\pm 0.032}$ | $0.554_{\pm 0.034}$ | $0.402_{\pm 0.032}$ | $0.466_{\pm 0.034}$ |
| o4-mini | $0.743_{\pm 0.027}$ | $0.859_{\pm 0.024}$ | $\underline{0.494}_{\pm 0.032}$ | $\underline{0.600}_{\pm 0.033}$ | $\mathbf{0.420}_{\pm 0.032}$ | $\underline{0.495}_{\pm 0.034}$ |

Table 1: VCI E-Score. For all metrics, higher is better (↑). **Best**, 2nd-best.

Our 8 models break down into closed vs. open-weight and reasoning vs. non-reasoning models. We use three closed-source, non-reasoning models: GPT-4o [59], GPT-4o-mini [59], and Claude Sonnet 3.7 (Sonnet 3.7) [60]. We use two open-weight, non-reasoning models: Qwen2.5 72B (Qwen2.5) [61] and Llama 4 Maverick (Llama 4) [62]. We use three total reasoning models, one open-weight—Deepseek R1 [63]—and two closed-source—o3-mini [64] and o4-mini [65].

## 5.1 VCI Evidence Scoring Benchmark

**Zero-shot benchmarking.** Table 1 shows performance across the three hierarchies of codes. Smaller models, such as GPT-4o-mini and Qwen-72b underperform on evidence scoring to larger models across every level. Second, we find that finer-grained codes are classified better by larger models and reasoning models. Reasoning models exhibit lower precision but higher recall on primary codes (i.e., higher-level classification of pathogenic or benign data), with reasoning models becoming better overall in precision for tertiary codes. However, overall performance on tertiary code prediction is poor, with the best model (o4-mini) getting only 0.420 Precision@5. The best-performing models on tertiary code prediction are reasoning models, and the best of the non-reasoning models are GPT-4o and Llama 4, two of the largest models included in this study. Tertiary code performance are also shown in Figure 1c.

**In-context prompting.** In addition to zero-shot models, we experiment with in-context example (ICL) prompting of GPT-4o, Llama-4, and o4-mini because of their superior zero-shot performance. In Figure 3, we show Precision@5 across primary, secondary, and tertiary codes for in-context examples (ICL Shot) of 5, 10, 20, and 30. With flat dotted lines, we show a reference 5-shot prompt with full-text included for each in-context example. We test with the description-only prompts for 5, 10, 20, and 30-shot examples, as described in Section 4.1. Secondary and tertiary codes follow an expected pattern for Llama-4 and o4-mini, increasing performance monotonically with more ICL shot, with smaller gains in performance from 20-shot to 30-shot. On primary code classification, ICL prompting improves o4-mini substantially, but these benefits do not transfer to GPT-4o and Llama-4. For GPT-4o, performance is best across all codes at 20-shot and then goes down at 30-shot. Finally, more in-context examples seems to equalize performance of the three models tested.

## 5.2 VCI Evidence Verification Benchmark

| Method | Evidence Verification | | | |
| | Positive Rate | TNR (↑) | TPR(↑) | F1(↑) |
|---|---|---|---|---|
| Random | $0.500_{\pm 0.032}$ | $0.500_{\pm 0.032}$ | $0.500_{\pm 0.032}$ | $0.500_{\pm 0.032}$ |
| GPT-4o-mini | $0.510_{\pm 0.032}$ | $\mathbf{0.602}_{\pm 0.043}$ | $0.655_{\pm 0.047}$ | $0.604_{\pm 0.038}$ |
| GPT-4o | $0.664_{\pm 0.030}$ | $0.443_{\pm 0.043}$ | $\mathbf{0.801}_{\pm 0.039}$ | $\mathbf{0.634}_{\pm 0.034}$ |
| Sonnet 3.7 | $0.660_{\pm 0.031}$ | $0.391_{\pm 0.042}$ | $0.722_{\pm 0.044}$ | $0.576_{\pm 0.036}$ |
| Qwen 2.5 72B | $0.502_{\pm 0.032}$ | $0.514_{\pm 0.043}$ | $0.525_{\pm 0.048}$ | $0.486_{\pm 0.042}$ |
| LLaMA 4 | $0.680_{\pm 0.031}$ | $0.377_{\pm 0.041}$ | $0.752_{\pm 0.043}$ | $0.587_{\pm 0.036}$ |
| Deepseek R1 | $0.672_{\pm 0.030}$ | $0.427_{\pm 0.042}$ | $0.799_{\pm 0.040}$ | $\underline{0.629}_{\pm 0.035}$ |
| o3-mini | $0.614_{\pm 0.031}$ | $0.492_{\pm 0.042}$ | $0.754_{\pm 0.042}$ | $0.625_{\pm 0.036}$ |
| o4-mini | $0.544_{\pm 0.032}$ | $0.480_{\pm 0.045}$ | $0.574_{\pm 0.047}$ | $0.509_{\pm 0.040}$ |

Table 2: VCI E-Ver. TNR = true negative rate; TPR = true positive rate. **Best**, 2nd-best.

**Zero-shot benchmarking.** With respect to overall performance (F1 score), we find that GPT-4o is the best-performing zero-shot model, with Deepseek-R1 being just 0.005 F1 points behind (Table 2). Across every LM tested, performance is poor for this task, with GPT-4o only exhibiting 0.634 F1 score; surprisingly, o4-mini and Qwen2.5 performance close to random on this task. However, we see that most models exhibit much better true positive rates (TPR), with higher TPR often correlating with better F1 score. Many models predict close to 2/3 of all samples as being "met" (i.e., Pos. Rate in Table 2) when the actual test set positive rate is 0.434. This indicates that models are overconfident in codes being met rather than not-met than the ground-truth data indicates. Thus, a major challenge for prompting LMs for the this task is determining whether sufficient evidence has been met within a paper to satisfy a code. Zero-shot F1 performance is shown in Figure 1d.

**In-context prompting.** We benchmark several ICL prompts for the evidence verification task across GPT-4o and Deepseek-R1. We interestingly find that ICL prompting, unlike in the VCI Escore task, *does not* correlate positively with the number of examples. The best-performing variant of GPT-4o prompting used 10 in-context examples at 0.657, whereas performance drops to 0.633 and 0.639 for 20 and 30 examples, respectively (Appendix Table 10). For Deepseek-R1, the best-performing variant is at 5-shot ICL with full-text in each example (0.641), but all description-only in-context prompting methods result in scores lower than zero-shot scores (Appendix Table 11). This indicates that the in-context examples might be less helpful for reasoning models in this task, and more exploration is needed to find a better prompting techniques to increases performance for reasoning models.

### 5.3 GCI Evidence Extraction Benchmark

In Table 3, results are presented on benchmarking of four LLMs on the GCI evidence extraction task. For category matching, performance is mixed across precision and recall metrics. GPT-4o achieves the highest precision rate at 0.493, meaning that almost 50% of predicted evidence categories match a ground-truth evidence category for the sample (Figure 1e). However, the highest recall is achieved by o4-mini at 0.835, but this is most likely due to the increased number of predicted evidence pieces in general for o4-mini, which is on average 3.397 as opposed to 2.307 for GPT-4o, 2.450 for Deepseek-R1,

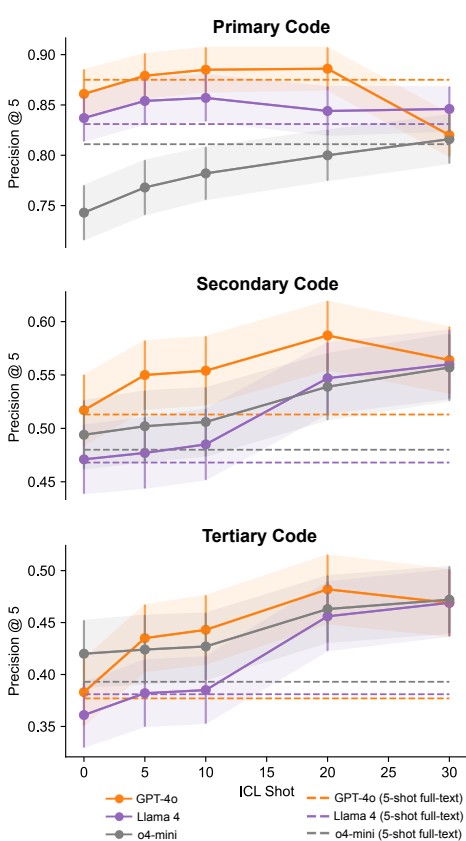

Figure 3: In-context prompting for VCI E-score task. Dotted lines are full-text 5-shot examples; shaded area represents standard error for each shot.

and 3.140 for Llama-4. All models predict more pieces of evidence than the ground-truth average: 1.273, around 1/2 of the number of evidence extractions predicted for each model. The gap between precision and recall might indicate that all models are extracting more evidence than was deemed important by GCI curators; this calls for development of evidence prioritization mechanisms, potentially in agentic setups or improved prompting.

| | **Category Matching** | | **Structure Adherence** | **Scoring** | |
| Method | Precision | Recall | Success | Normalized MAE ($\downarrow$) | $\Delta$Strength ($\uparrow$) |
|---|---|---|---|---|---|
| Random | $0.063_{\pm0.001}$ | $0.126_{\pm0.002}$ | N/A | $0.304_{\pm0.001}$ | $0.158_{\pm0.002}$ |
| GPT-4o | $\mathbf{0.493}_{\pm0.017}$ | $\underline{0.787}_{\pm0.020}$ | $\underline{98.81\%}$ | $\underline{0.196}_{\pm0.009}$ | $0.342_{\pm0.027}$ |
| Llama 4 | $0.363_{\pm0.013}$ | $\underline{0.787}_{\pm0.020}$ | $\mathbf{99.40\%}$ | $0.393_{\pm0.120}$ | $0.129_{\pm0.019}$ |
| Deepseek R1 | $\underline{0.456}_{\pm0.022}$ | $0.734_{\pm0.028}$ | $61.61\%$ | $0.228_{\pm0.015}$ | $\underline{0.346}_{\pm0.035}$ |
| o4-mini | $0.425_{\pm0.015}$ | $\mathbf{0.835}_{\pm0.019}$ | $96.73\%$ | $\mathbf{0.186}_{\pm0.010}$ | $\mathbf{0.445}_{\pm0.025}$ |

Table 3: GCI task: experimental evidence extraction. **Best**, 2nd-best.

All models adhere to structure at over 95% except for Deepseek R1, which breaks the desired structure 40% of the time. In scoring, the best overall performance is achieved by o4-mini, indicating that it is able to better discern the strength of the evidence. It achieves a normalized MAE of 0.186, which is close to the performance of GPT-4o; however, o4-mini's captures $\Delta$Strength significantly better than GPT-4o, as does DeepSeek-R1. Of note is the relative weak performance across all models on this task. o4-mini, the best-performing model, only matches the strength change less than 50% of the time, meaning that its ability to determine if evidence is better or worse than a default score is roughly random. This highlights the need to design LMs methods that can align LMs to interpret evidence similar to the ways in which humans interpret evidence as out-of-the-box, these LMs are poor at this task.

### 5.4 LLM-as-a-judge for evaluating LM explanations vs. ClinGen explanations

We first perform a small-scale correlation against a manually-reviewed set of VCI vs. LM explanations. This reveals that the **task-aware** method performs best, with a 0.744 F1 correspondence to the manual matching; evidence-aware performs slightly behind at 0.723 F1 and task-agnostic performs worse at 0.615 F1. Thus, we use the task-aware judge method to measure explanation quality. More details are given in Appendix I.

**LM judge results.** We then apply the LM judge technique to measure the concordance of LM explanations to ClinGen explanations for correctly-classified VCI E-score examples. We test three models—GPT-4o, Llama 4, and o4-mini—along with their zero-shot and 30-shot prompted variants. GPT-4o 30-shot overall performs the best with 70.4% agreement with the VCI explanation; not far behind is o4-mini, which at 30-shot agrees at a 68.3% rate. Interestingly, in-context examples improve explanation performance across the board. All models show at least some boost agreement when applying in-

| Model | Judge Agreement |
|---|---|
| GPT-4o (zero-shot) | $0.486_{\pm 0.031}$ |
| GPT-4o (30-shot) | $\mathbf{0.704}_{\pm 0.025}$ |
| Llama-4 (zero-shot) | $0.534_{\pm 0.043}$ |
| Llama-4 (30-shot) | $0.564_{\pm 0.034}$ |
| o4-mini (zero-shot) | $0.657_{\pm 0.040}$ |
| o4-mini (30-shot) | $\underline{0.683}_{\pm 0.032}$ |

Table 4: LM-as-judge results for VCI E-Score models on zero-shot and 30-shot prompting.

context prompting, but especially GPT-4o, which jumps 44.8% in agreement level from zero-shot. A hypothesis for this phenomenon is that in-context examples allow the models to match the style of the explanations in the VCI, increasing their likelihood to be classified as similar by the judge.

## 6 Discussion

CGBENCH presents a unique benchmark to assess real-world scientific literature evaluation of LMs for scientific tasks, such as genetic variant curation and classification. CGBENCH raises a number of unique and novel problems based on the difficulty of its tasks: How can LM performance be improved on highly-specialized, precise instructions? How can LM outputs be aligned to how humans interpret scientific evidence? Are current systems fundamentally misaligned on this task, or are there ways to easily adapt them? Such challenges emerge when considering complex, real-world settings that more closely mimic scientific practice and translation. We hope that given the coverage of tasks and size of CGBENCH, especially in the GCI, it can be used to explore prompt-tuning [66] or memory approaches [67] that can encode large numbers of demonstrations or general rules based on observing many VCI/GCI entries. Further, we hope the inclusion of expert explanations from ClinGen will spur work into evaluating LMs in science on free-text explanations rather than solely on structured classification frameworks. The Appendix contains a discussion of broader impacts (Appendix A) and limitations (Appendix B). CGBENCH sets a new standard in complex scientific literature interpretation and opens new challenges for modern LMs to drive future development at the intersection of AI and scientific research.

## Acknowledgements

We thank members of the Zou lab, especially Jake Silberg and Kyle Swanson, for feedback on the work. O.Q. and H.Z. are supported by graduate fellowship awards from Knight-Hennessy Scholars at Stanford University.

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

## Appendix A    Broader Impacts Statement

This work carries broader implications for both AI research and clinical genomics. By introducing CGBENCH as a benchmark for evidence-driven gene and variant curation, we push the development of LMs towards complex, multi-step scientific reasoning. This work demonstrates that LMs still struggle in precisely-defined scientific extraction tasks, and it puts forward a task that is more complex than previous question-answering benchmarks on scientific publications. We hope that this work spurs discussion and broad consideration in the community for how LMs can be prompted and evaluated for guideline-following that holds up to rigorous standards required for clinical research and other high-stakes research areas. While only limited to clinical genetics, this work contributes to a broad body of research in probing LM capabilities in science, and we hope it also inspires other benchmarks for complex tasks in a variety of scientific fields. Specific to the tasks presented here, CGBENCH also concerns a task of great importance to clinical research. Clinical gene and variant curation are vital for translation of genetic data to clinical insights; accelerating review of evidence for these tasks could advance personalized medicine, benefiting the lives of patients around the world.

## Appendix B    Limitations

A few weaknesses and limitations of CGBENCH are worth noting. One particular weakness of CGBENCH is omitting supplementary information from the context available to LMs. It has been estimated that much of the evidence for variants can be found in the supplementary sections of papers [S1, S2]. However, including supplementary data is challenging for many LMs as much of supplementary data is included in multimodal, structured data such as tables or figures that are not as easily processed by standard LM prompting approaches. In a similar vein, valuable information may be encoded in multimodal data included in scientific publications such as tables and figures, but this is also not included in CGBENCH. In addition, some codes in the ERepo require multiple publications to meet the criteria, and a point of future work would be to aggregate multiple papers to evaluate multi-document reasoning for codes requiring multiple studies.

The LM judge implementation has a number of limitations; more extensive validation with either user studies or broad analysis would be needed to understand the trustworthiness of such a method. Finally, while CGBENCH is a rigorous dataset for clinical genetics, there may exist certain biases that are specific to clinical genetics as a field. For example, [44] highlights that variants in particular are not always referenced by the same nomenclature across publications. Thus, some difficulty in benchmarking tasks could be due to poor interpretation of variant names and lack of translation to references in-text. Gene names have a similar phenomenon [S4]. These specialized problems may hinder broad conclusions about performance of LMs on scientific data, but quantifying these difficulties in benchmarking is a highly non-trivial task.

## Appendix C    Further discussion of background

**Similar prediction settings.** None of the related works mentioned use exactly the settings presented in CGBENCH. The closest to the CGBENCH VCI evidence scoring is perhaps LitGen [50], which uses a long short-term memory (LSTM) network to classifying the type of evidence presented in a paper for a given variant. This type of approach is limited in its ability to transfer to diverse VCEP codes and adapt to tasks outside of evidence scoring (both ideas discussed in Section 3). In contrast, we focus on the capabilities of generative LMs to perform this task, given the ability to integrate these models into modern agentic systems and to vary nuanced instructions for evolving guidelines.

**Variant effect prediction and similarities.** Many works have studied the general problem of variant effect prediction, which is primarily focused on utilizing biomolecular data, such as genomic or protein-level sequences [S6, S7, S8, S9, S10]. Others have approached the variant effect prediction task by classifying text related to that variant [S11], and other works have attempted to perform gene-disease association prediction [S4, S12, S13]. Datasets have been developed for benchmarking variant effect predictors [S14]. However, CGBENCH distinctly focuses on the task of evidence scoring for publications related to variants rather than predicting the effect of novel mutations on protein or genomic sequences.

**Additional context/information for VCI.** The variant curation interface (VCI) is a rich resource for variant classification that supports the application of the Richards et al. ACMG/AMP guidelines [8]. Variant can be classified as "pathogenic", e.g., variants that increases an individual's susceptibility or predisposition to a certain disease or disorder [S15], or "benign", e.g., variants that are not associated with an abnormal phenotype or increased disease risk [S16] (please see [S17] for a discussion the full complexity of this terminology). The goal of a variant curation, as defined in original ACMG/AMP guidelines [8], is to classify a variant annotation—which consists of a variant within a given gene, a disease, and a mode of inheritance—to either "Pathogenic", "Likely Pathogenic", "Variant of Uncertain Significance", "Likely Benign", or "Benign" based on the available evidence that is gathered by curators.

In the E-Score task, samples per VCEP range from 1 to 41 with a median of 5. In the E-Ver task, the range is from 1 to 68 with a median of 5. VCEPs with large numbers of samples are often those with well-studied genetic bases for disease, such as Phenylketonuria and RASopathy. Several VCEPs have sample sizes of only 1; however, this does not invalidate such samples as the transfer across VCEPs is reasonable given the similarity in guidelines.

Note that in this work while we treat the VCEP guidelines as separate entities, these guidelines are in practice very similar. The ACMG/AMP protocol (Richards et al. [8]) serves as the basis for how a VCEP determines their guidelines, and often only a few codes are changed within the base guidelines provided by Richards et al. This is evidenced by the high similarity in guidelines across the different VCEPs. Defining similar VCEP guidelines as those with a ROUGE-L score >0.95, we find that 62% of the codes have over 75% reuse across all VCEP guidelines, meaning that 75% of VCEPs that use such a code decide to reuse a standard definition. This high similarity across VCEPs illustrates the similarity of the variant curation task, allowing us to draw effective conclusions even given a small number of VCEPs for some samples.

**Additional context/information for GCI.** Like the VCI, the gene curation interface (GCI) is a platform that supports the application of rigorous guidelines for determining gene-disease validity [23]. The current standards of practice (SOP) have been published at [S19]. Please reference this SOP for full details on gene-disease validity curation as the exact details are comprehensive and out-of-scope for this work. In brief, determining a gene-disease relationship comes down to aggregating and scoring evidence. With some additional heuristics, all evidence and its scores are summed together to comprise the final score. The scores predicted, as described in Section 3.2, are part of this cumulative score. However, the experimental evidence, as we focus on in this paper, is just one part of the analysis procedure. The case-level segregation and case-level variants analyses are left as future work as these elements are not as easily framed as queries to LMs.

**Size of CGBENCH compared to other benchmarks.** The size of CGBENCH is comparable to similar human-written (e.g., not automatically mined or generated) benchmarks of scientific capabilities of language models. LitQA2 [10], a question-answering benchmark on scientific papers, is 248 samples and GPQA [S21], a standard benchmark in challenging scientific tasks, is 448 samples, with GPQA-Diamond, the most challenging set, being 198 samples. This puts CGBENCH's VCI tasks as comparably-sized and the GCI task as much larger, at 2,155 samples.

## Appendix D   Data preparation and cleaning

### D.1   Data filtering

**Filtering for evidence-sourced samples.** In filtering for the VCI, we scraped all annotations of evidence codes from the ERepo that contain references to PubMed IDs (PMIDs). Many code annotations omit PubMed IDs, especially those that focus on population-level statistics taken from databases. We then filter to only samples that cite open-source papers where both full-text and abstracts are accessible. However, of the samples processed from ERepo, only roughly 20% of the papers cited were available via full-text. More discussion on this is given in Appendix D.

**Open-source publications.** We filter PMIDs that only have full-text. We show in Figure 4 that only roughly 20% of the papers in the VCI and 23% of papers in the GCI have full-text entries that are publicly available for automatic scraping. The main issues when scraping were 1) lack of PubMed

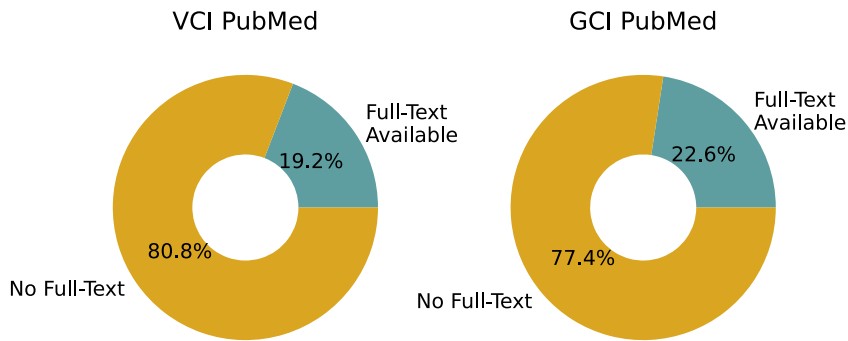

Figure 4: Results of attempting to scrape full-text results from VCI and GCI.

Central (PMC) IDs, i.e., no openly-available full-text through PubMed, and 2) publishers not allowing XML scraping. Thus, we only stuck to papers available through PMC.

**Deduplication of evidence verification.** The evidence verification (E-Ver) task requires determining whether a code is met based on the presented evidence. Many times in the VCI, multiple PubMed IDs are used to determine if a code is "met". We filter out the examples where multiple PubMed IDs are associated to one code being met or not met. The rationale behind this is twofold: 1) we want to ensure all necessary evidence is included in the context of the model when prompting. This is achieved by narrowing to samples where only one PubMed ID is necessary to make the evidence determination. 2) we want to avoid scenarios where some information is contained a publicly-available PubMed paper and other information is not (i.e., is behind a paywall). Since a large number of PubMed IDs cited on the VCI do not have full-text available to scrape, we therefore stick to only samples where one PubMed ID that is openly-available is associated to the evidence determination. One point of future work would be multi-document evidence verification, but this is left for follow-up studies.

## D.2   Curation of VCEP Criteria Specifications

**Further explanation of Variant Curation Expert Panels (VCEPs) and evidence codes.** Each entry in the VCI is supervised by a variant curation expert panel (VCEP). This is a group of domain experts that supervises the curation of variants for a given group of genes and/or diseases. VCEPs develop their own guidelines for their specific areas, modifying the specification of evidence codes through criteria specifications in an interface known as the CSpec. We process all VCEP criteria specifications in the CSpec for the samples used in CGBENCH and provide automatic loading mechanisms for these criteria in prompts for VCI tasks (discussed in Section 4).

VCEP evidence codes use a wide variety of evidence to determine pathogenicity or benignity of a variant, but in CGBENCH, we stick to that available through published papers. Thus, we only curate code annotations that include an associated PubMed ID. However, to measure the ability of LMs to discern literature-related codes from non literature-related codes, we provide the full breadth of codes as the context for the prompts (see Section 4.1 for more details).

The original ACMG/AMP guidelines [8] lays out a procedure through which variants are classified, and this procedure utilizes evidence codes at its core. In brief, the goal of the VCI is to classify a variant, disease, and mode of inheritance tuple as "pathogenic", "likely pathogenic", "variant of uncertain significance", "likely benign", or "benign". The "likely" qualifier means that the classification is less certain that if "likely" is omitted. "Variant of uncertain significance" (VUS) means that the variant does not have enough data to classify as pathogenic or benign, or that the data is contradictory and requires more study. To make these classifications, each VCEP determines a set of heuristics that combines the evidence codes into a final determination. Some evidence codes are weighed higher than others, e.g., "strong" codes are higher than "supporting" codes. We do not delve into the details here because this is out of scope and an active research area in clinical genetics, but please reference [8] for information on the overall guidelines for this procedure.

**Scraping VCEPs.** We put significant effort into scraping `clinicalgenome.org` to get the specific VCEPs used for each example. This involved several important considerations. First, each VCEP defines a set of criteria for specific genes and diseases, so we controlled for this when creating prompts. Second, VCEPs periodically publish updates to their guidelines, but the guidelines used to curate each example are not available for most VCI entries. Thus, we infer the guidelines used for classification by taking a max-min on the date. For each example, we look at the published date and take the VCEP version that is the newest (max) but below the published date of the example. Third, VCEP guidelines are inconsistent in denoting which codes are available and which are not. We confirmed that codes with quaternary qualifiers (e.g., "_Supporting", "_Strong", etc.) that are labeled as "not applicable" are not used for classification. However, codes without quaternary qualifiers, such as "BS1" or "PS2", may be used for classification even if labeled as "not applicable". Therefore, we choose to include all tertiary codes for each VCEP in the options for the prompt to the LM. It is unclear whether these discrepancies are intended or are errors in the VCI entries.

### D.3 Additional GCI Curation Information

**Gene curation standard of practice (SOP).** The analog to VCEPs for gene curation are the standards of practice (SOPs) published by ClinGen. SOPs are consistent across all gene-disease curations and are updated globally every few years. We directly gathered all SOPs that were used to classify current gene-disease associations. When constructing prompts for the GCI, we use the appropriate SOP used to classify that gene-disease entry.

**GCI experimental categories.** As mentioned in Section 3.3, there are 13 unique experimental categories included for the GCI across all samples. These 13 experimental categories are consistent across all SOPs, with slightly different descriptions across each of them. The full list of experimental categories is:

1. Biochemical Function A
2. Biochemical Function B
3. Protein Interaction
4. Expression A
5. Expression B
6. Functional Alteration Patient cells
7. Functional Alteration Non-patient cells
8. Model System Non-human model organism
9. Model Systems Cell culture model
10. Rescue Human
11. Rescue Patient Cells
12. Rescue Non-human model organism
13. Rescue Cell culture model

There are six main groups of categories: biochemical function, protein interaction, expression, functional alteration, model systems, and rescue, which are the prefixes to the above categories. Some groups differentiate subtypes of these categories, such as "expression A" and "expression B", which are differentiated by if the method used was to detect RNA transcripts (expression A) or protein expression (expression B). Thus, models must be able to differentiate between these methodological approaches when categorizing evidence. Full descriptions for each category across SOPs are given in our dataset.

**GCI evidence score.** $s_i$ is the score such that $s_i \in [b_l^{\text{sop}}, b_u^{\text{sop}}]$ where $b_l^{\text{sop}}, b_u^{\text{sop}} \in \mathbb{R}_{\geq 0}$ are lower and upper bounds (respectively) defined by the SOP being used to categorize the query. These bounds are given to the LM in the input prompt; we do not perform additional filtering to ensure they are met during processing of outputs as we assume the bounds are followed by the LM.

# Appendix E    In-context prompting full results - VCI Evidence Scoring

| | Evidence Scoring | | | | | |
| | Primary | | Secondary | | Tertiary | |
| Method | Precision@5 | Recall@5 | Precision@5 | Recall@5 | Precision@5 | Recall@5 |
|---|---|---|---|---|---|---|
| GPT-4o (zero-shot) | $0.861 \pm 0.024$ | $0.878 \pm 0.023$ | $0.517 \pm 0.033$ | $0.568 \pm 0.033$ | $0.383 \pm 0.032$ | $0.427 \pm 0.033$ |
| GPT-4o (5-shot, FT) | $0.875 \pm 0.022$ | $0.888 \pm 0.022$ | $0.513 \pm 0.032$ | $0.573 \pm 0.033$ | $0.377 \pm 0.032$ | $0.437 \pm 0.034$ |
| GPT-4o (5-shot) | $0.879 \pm 0.022$ | $0.888 \pm 0.022$ | $0.550 \pm 0.032$ | $0.610 \pm 0.033$ | $0.435 \pm 0.032$ | $0.476 \pm 0.033$ |
| GPT-4o (10-shot) | $0.885 \pm 0.022$ | $0.898 \pm 0.021$ | $0.554 \pm 0.032$ | $0.624 \pm 0.033$ | $0.443 \pm 0.033$ | $0.478 \pm 0.033$ |
| GPT-4o (20-shot) | $0.886 \pm 0.021$ | $0.898 \pm 0.021$ | $0.587 \pm 0.032$ | $0.617 \pm 0.033$ | $0.482 \pm 0.033$ | $0.488 \pm 0.033$ |
| GPT-4o (30-shot) | $0.820 \pm 0.021$ | $0.902 \pm 0.021$ | $0.564 \pm 0.031$ | $0.639 \pm 0.032$ | $0.469 \pm 0.032$ | $0.532 \pm 0.034$ |
| Llama 4 (zero-shot) | $0.837 \pm 0.023$ | $0.873 \pm 0.023$ | $0.471 \pm 0.032$ | $0.532 \pm 0.034$ | $0.361 \pm 0.031$ | $0.424 \pm 0.034$ |
| Llama 4 (5-shot, FT) | $0.831 \pm 0.024$ | $0.883 \pm 0.022$ | $0.468 \pm 0.032$ | $0.539 \pm 0.034$ | $0.381 \pm 0.032$ | $0.427 \pm 0.033$ |
| Llama 4 (5-shot) | $0.854 \pm 0.023$ | $0.883 \pm 0.022$ | $0.477 \pm 0.033$ | $0.507 \pm 0.034$ | $0.382 \pm 0.032$ | $0.410 \pm 0.033$ |
| Llama 4 (10-shot) | $0.857 \pm 0.023$ | $0.883 \pm 0.022$ | $0.485 \pm 0.033$ | $0.544 \pm 0.034$ | $0.385 \pm 0.032$ | $0.429 \pm 0.034$ |
| Llama 4 (20-shot) | $0.844 \pm 0.024$ | $0.878 \pm 0.023$ | $0.547 \pm 0.033$ | $0.593 \pm 0.033$ | $0.456 \pm 0.033$ | $0.490 \pm 0.034$ |
| Llama 4 (30-shot) | $0.846 \pm 0.022$ | $0.907 \pm 0.020$ | $0.560 \pm 0.032$ | $0.632 \pm 0.033$ | $0.469 \pm 0.032$ | $0.534 \pm 0.034$ |
| o4-mini (zero-shot) | $0.743 \pm 0.027$ | $0.859 \pm 0.024$ | $0.494 \pm 0.032$ | $0.600 \pm 0.033$ | $0.420 \pm 0.032$ | $0.495 \pm 0.034$ |
| o4-mini (5-shot, FT) | $0.811 \pm 0.023$ | $0.917 \pm 0.019$ | $0.480 \pm 0.031$ | $0.622 \pm 0.033$ | $0.393 \pm 0.030$ | $0.505 \pm 0.034$ |
| o4-mini (5-shot) | $0.768 \pm 0.027$ | $0.844 \pm 0.025$ | $0.502 \pm 0.033$ | $0.561 \pm 0.034$ | $0.424 \pm 0.033$ | $0.463 \pm 0.034$ |
| o4-mini (10-shot) | $0.782 \pm 0.026$ | $0.873 \pm 0.023$ | $0.506 \pm 0.032$ | $0.595 \pm 0.033$ | $0.427 \pm 0.032$ | $0.490 \pm 0.034$ |
| o4-mini (20-shot) | $0.800 \pm 0.025$ | $0.878 \pm 0.023$ | $0.539 \pm 0.031$ | $0.649 \pm 0.032$ | $0.463 \pm 0.032$ | $0.546 \pm 0.034$ |
| o4-mini (30-shot) | $0.816 \pm 0.024$ | $0.898 \pm 0.021$ | $0.557 \pm 0.031$ | $0.661 \pm 0.032$ | $0.472 \pm 0.032$ | $0.566 \pm 0.033$ |

Table 5: VCI E-Score performance in tabular form. Pairs with results in Figure 3. FT = "full-text". If "FT" is not mentioned, it is assumed to be description-only prompting.

This section contains full results from experiments in the main text. Table 5 shows results from the in-context demonstrations experiment detailed in main-text Figure 3.

| | Evidence Scoring (Tertiary) | |
| **Method** | Precision@5 | Recall@5 |
|---|---|---|
| S1 | 0.482 | 0.488 |
| S2 | 0.492 | 0.500 |
| S3 | 0.482 | 0.527 |
| S4 | 0.494 | 0.505 |
| S5 | 0.473 | 0.512 |

Table 6: VCI E-Score results for GPT-4o with multiple seeds (S*N*) used to sample in-context examples. Results shown are average across tertiary code scoring.

As an exploration into the stability of choice of in-context examples, we ran multiple sets of in-context examples when building prompts for the VCI E-score task. In the paper, we use one static choice of examples for consistency, but in this case, we vary the examples chosen. We obtain 5 different combinations of examples for the in-context prompting and show the results in Table 6. Results show surprisingly stable results across different choices of seeds, showing that the choice of in-context examples might not have significant effect on performance.

## Appendix F    Additional Experiments

### F.1    Summarization for ICL Compression

We discuss in Section 4.1 about the tradeoffs between prompting in-context examples with full-text vs. descriptions only. During development of CGBENCH, we also experimented with an agentic RAG-like system which first summarizes key findings in in-context papers with a smaller LM (GPT-4o-mini) and then uses these in the in-context examples in place of the full paper. The results from this experiment can be found in Table 7 for the VCI E-score task. We experiment with replacing in summarized text for both in-context examples and the actual input for the sample. Overall, the summarization does not substantially change performance when compared to prompting with no

| Method | Evidence Scoring | | | | | |
| | Primary | | Secondary | | Tertiary | |
| | Precision@5 | Recall@5 | Precision@5 | Recall@5 | Precision@5 | Recall@5 |
|---|---|---|---|---|---|---|
| GPT-4o (zero-shot, f) | 0.861±0.024 | 0.878±0.023 | 0.517±0.033 | 0.568±0.033 | 0.383±0.032 | 0.427±0.033 |
| GPT-4o (zero-shot, s) | 0.880±0.022 | 0.898±0.021 | 0.508±0.033 | 0.573±0.034 | 0.387±0.032 | 0.424±0.033 |
| GPT-4o (10-shot, s) | 0.888±0.022 | 0.893±0.022 | 0.543±0.033 | 0.585±0.033 | 0.425±0.033 | 0.449±0.033 |
| GPT-4o (10-shot, s+f) | 0.881±0.022 | 0.898±0.021 | 0.531±0.032 | 0.627±0.033 | 0.387±0.032 | 0.468±0.034 |
| Llama 4 (zero-shot, f) | 0.837±0.023 | 0.873±0.023 | 0.471±0.032 | 0.532±0.034 | 0.361±0.031 | 0.424±0.034 |
| Llama 4 (zero-shot, s) | 0.822±0.023 | 0.893±0.022 | 0.453±0.033 | 0.498±0.034 | 0.364±0.032 | 0.393±0.033 |
| Llama 4 (10-shot, s) | 0.873±0.023 | 0.883±0.022 | 0.482±0.034 | 0.480±0.033 | 0.352±0.033 | 0.363±0.032 |
| Llama 4 (10-shot, s+f) | 0.827±0.026 | 0.844±0.025 | 0.445±0.034 | 0.456±0.034 | 0.332±0.032 | 0.332±0.032 |

Table 7: VCI evidence scoring with summarization. "f" = full text; "s" = summarized text only; "s+f" = summarized text for in-context examples but full-text for actual input.

summarization. For GPT-4o, the performance is almost equivalent, with a change of -0.004 in primary Precision@5, +0.003 in secondary Precision@5, and -0.010 in tertiary Precision@5 for 10-shot with no summary vs. 10-shot with the summary. For Llama 4, performance drops for all levels compared to the non-summary 10-shot baseline.

## F.2 Correlating model outputs

Since we test a variety of models in this work, we can ask the question: how well do model predictions correlate across different models? Answering this might illuminate if the benchmark contains some examples which are poorly predicted by all models or if the models excel at different types of inputs.

For this analysis, we look at the VCI E-Score task and see which models correlate the most across predictions. Specifically, we're interested in zero-shot models and predictions on tertiary codes. Aligning predictions along examples, we compute a correlation matrix of each model, where the "correlation" is derived by taking the number of predictions in which the majority vote across 5 samples was equivalent across two models for a given example. This result is shown in Figure 5, where we see that models do not correlate significantly across predictions. Most models lie in the 0.4-0.6 range for correlations, with the exception of all the reasoning models, which have very high correlations. This hints at the fact that the reasoning models might be making systematic errors in their predictions. GPT-4o-mini and Claude Sonnet 3.7 seem to correlate the least with other models overall, while many models correlate higher in general with GPT-4o. These results point to the conclusion that model predictions don't overlap significantly, opening further room for investigation.

We then look into the binary correlation of if the majority vote answer was correct on the prediction task. In this case, we correlate predictions by a binary vector set to 0 if the answer is not in the ground-truth and 1 if the answer is in the ground-truth. Figure 6 shows the results of this analysis; most LMs correlate quite highly when measuring which predictions were correct or incorrect. The lowest correlation is 0.85 while the highest goes up to 0.96. This metric is highly correlated with "correctness" and the most-correct models tend to correlate the most (R1, o4-mini, and GPT-4o). The discrepancy in these results and the correlation in raw prediction indicates that for samples in which models are incorrect, they tend to make very different predictions, i.e., the models miss in different ways. These results also mean that there are a significant number of samples in which no language model is correct; in fact, 74.6% of samples in the VCI E-Score task have no LM that gets them correct. These samples are worth investigating for future work as they seem to be the most difficult.

## F.3 Test set contamination discussion and analysis

CGBENCH is designed to minimize contamination of the samples in three main ways. First, the ClinGen database's structure does not inject the full-text of the PubMed article beside the evidence classification for each sample. Only the PubMed ID is shown as a reference. Please see the ClinGen website `clinicalgenome.org` for more information on this. Second, no definition of the evidence code is shown on the ClinGen webpages. These evidence codes are defined in a separate database within the ClinGen site, so it is unlikely that LMs would associate these samples during large-scale

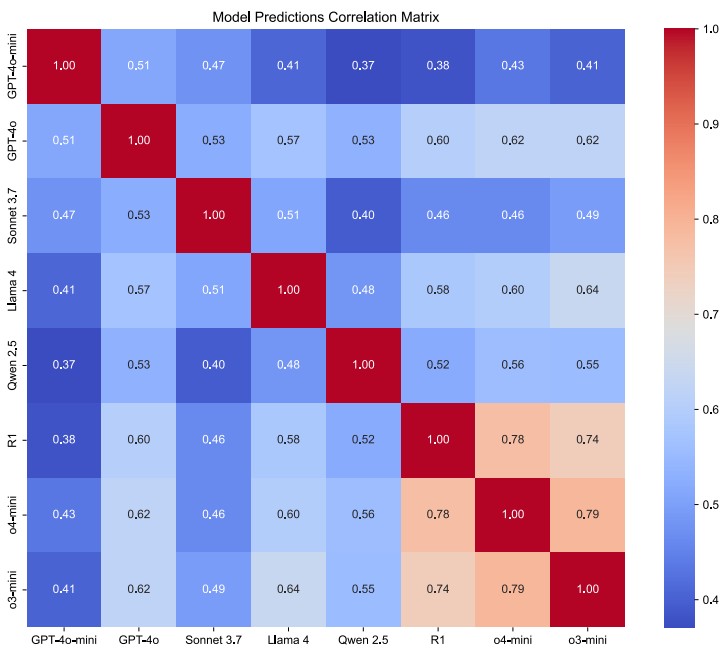

Figure 5: Prediction correlation for VCI E-Score across zero-shot models. All codes compared are tertiary codes.

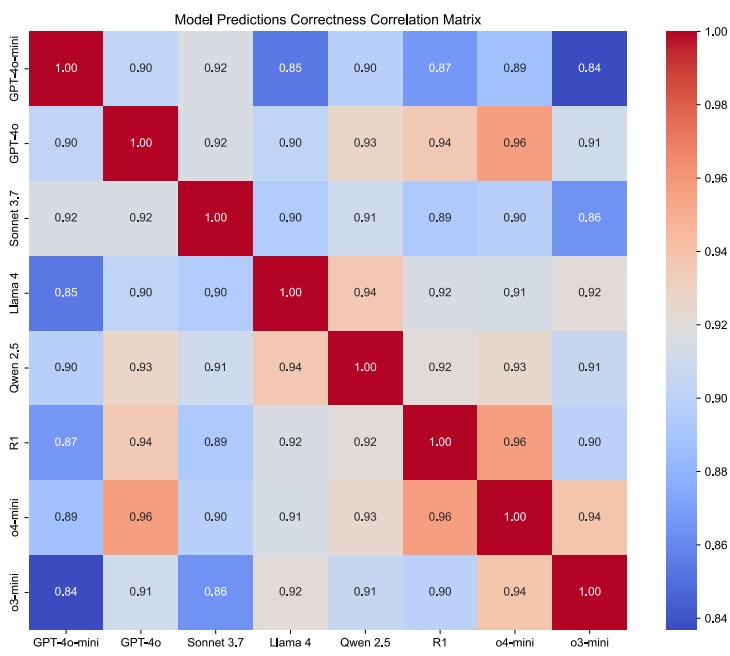

Figure 6: Correctness correlation for VCI E-Score across zero-shot models.

pretraining. Third, we design custom prompting for each example in the ClinGen database; since our prompts are custom-designed for the tasks we seek to benchmark, we are confident the models have not seen such prompts at training time.

One way to test model memorization is by using the PubMed IDs (PMIDs) in place of the papers. In this experiment, we replace the full-text of the paper with the PMID in the original prompt for the VCI E-Score task. This allows us to test if the language model is utilizing the paper's information to make it's determination or some other correlate, such as the variant or a memorized structure of the webpage. In another experiment (Random PMID), we replace the PMID with a random PMID, observing if the model is using the PMID to relate the variant/disease to pathogenicity or benignicity.

| Method | Evidence Scoring | | | | | |
| | Primary | | Secondary | | Tertiary | |
| | P@5 | R@5 | P@5 | R@5 | P@5 | R@5 |
|---|---|---|---|---|---|---|
| GPT-4o (PMID) | 0.841 | 0.858 | 0.361 | 0.505 | 0.152 | 0.277 |
| GPT-4o (Random PMID) | 0.462 | 0.652 | 0.161 | 0.328 | 0.055 | 0.125 |
| LLaMA 4 (PMID) | 0.754 | 0.887 | 0.343 | 0.473 | 0.189 | 0.294 |
| LLaMA 4 (Random PMID) | 0.767 | 0.877 | 0.327 | 0.475 | 0.138 | 0.260 |
| o4-mini (PMID) | 0.876 | 0.912 | 0.429 | 0.537 | 0.238 | 0.373 |
| o4-mini (Random PMID) | 0.851 | 0.917 | 0.342 | 0.500 | 0.196 | 0.331 |

Table 8: Experiment with PMIDs substituted for papers in the input prompts. Random PMID refers to a random PMID used in the prompt, i.e., not the PMID used for the classification. Compare to setup and results in Table 1

.

From these results, a few important trends emerge (see Table 8). First, there is a drop in performance for tertiary code prediction, which is the most fine-grained task in CGBENCH, e.g., requiring the most precise reasoning about the evidence. While not random, these results suggest that the model is relying on the content of the papers to make its determination about the code classification. Second, the drop in performance is less evident for Primary and Secondary codes. This is potentially because the model tries to assign pathogenic or benign labels to the variants without considering the evidence. This is reasonable given that the model may have prior knowledge about the variant's classification obtained during broad training on the literature, and it is thus answering based on the known classification of the variant. These results tell us that primary and secondary code tasks may contain some correlates that impact LM classification, but tertiary code classification requires comprehension of the papers given to the model.

## F.4 Training cutoff date analysis

| Method | Evidence Scoring (Tertiary) | |
| | Precision@5 | Recall@5 |
|---|---|---|
| Random (baseline) | $0.038_{\pm 0.083}$ | $0.180_{\pm 0.384}$ |
| GPT-4o (B) | $0.393_{\pm 0.033}$ | $0.439_{\pm 0.035}$ |
| GPT-4o (A) | $0.267_{\pm 0.114}$ | $0.267_{\pm 0.114}$ |
| LLaMA 4 (B) | $0.368_{\pm 0.032}$ | $0.437_{\pm 0.035}$ |
| LLaMA 4 (A) | $0.267_{\pm 0.114}$ | $0.267_{\pm 0.114}$ |
| o4-mini (B) | $0.431_{\pm 0.034}$ | $0.487_{\pm 0.035}$ |
| o4-mini (A) | $0.280_{\pm 0.090}$ | $0.600_{\pm 0.126}$ |

Table 9: Performance of different models on the tertiary code prediction task for E-Score before (B) and after (A) the Claude Sonnet 3.7 cutoff date. We subset to variant curations published before and after the cutoff date (November 2024).

We examine the performance on samples published after the model cutoff dates vs. those published before. For consistent comparison, we use the latest knowledge cutoff date of all the models in this group which is Claude Sonnet 3.7 with a cutoff date of November 2024 [3]. This allows us to compare all models across the same samples that are after their respective knowledge cutoff dates. We split results into those before the cutoff date and after the cutoff date.

---

[3]https://docs.anthropic.com/en/docs/about-claude/models/overview

These results in Table 9 show a drop in performance across all models from before the cutoff to after the cutoff, but this drop is modest. Such date cutoff ability is a benefit of CGBENCH, allowing users to audit the effect of timescale for when a curation was published against LM performance.

## Appendix G  In-context prompting full results - VCI Evidence Verification

This section shows the full results that are referenced in main-text for the in-context demonstration experiments on GPT-4o (Table 10) and DeepSeek-R1 (Table 11) for the VCI E-Ver task.

| GPT-4o Method | Evidence Verification (F1@1 $\uparrow$) |
|---|---|
| Original | $0.634_{\pm 0.034}$ |
| ICL 5 - FT | $0.617_{\pm 0.035}$ |
| ICL 5 - Desc only | $0.637_{\pm 0.035}$ |
| ICL 5 - Summarize + FT | $0.621_{\pm 0.034}$ |
| ICL 10 - Desc only | $\mathbf{0.657}_{\pm 0.033}$ |
| ICL 10 - Summarize + FT | $0.633_{\pm 0.037}$ |
| ICL 20 - Desc only | $0.633_{\pm 0.035}$ |
| ICL 30 - Desc only | $0.639_{\pm 0.036}$ |

Table 10: VCI task — evidence-sufficiency metric with augmentations

| Deepseek-R1 Method | Evidence Verification (F1@1 $\uparrow$) |
|---|---|
| Original | $0.628_{\pm 0.034}$ |
| ICL 5 - FT | $\mathbf{0.641}_{\pm 0.034}$ |
| ICL 5 - Desc only | $0.597_{\pm 0.037}$ |
| ICL 10 - Desc only | $0.611_{\pm 0.039}$ |
| ICL 20 - Desc only | $0.624_{\pm 0.036}$ |
| ICL 30 - Desc only | $0.621_{\pm 0.034}$ |

Table 11: VCI task — evidence-sufficiency metric with augmentations

| Seed | Evidence Verification (F1@1 $\uparrow$) |
|---|---|
| S1 | 0.657 |
| S2 | 0.655 |
| S3 | 0.651 |
| S4 | 0.652 |
| S5 | 0.658 |

Table 12: Stability results on GPT-4o for choice of in-context examples across multiple seeds (S$N$).

Table 12 shows results for using multiple seeds for choosing in-context examples for the E-Ver task. Results show significant stability across choices of in-context examples, opening the potential to future research beyond in-context example choice.

## Appendix H  LM judge details

### H.1  Judge setup

We formalize the judge setup as a prediction between two explanations, $e_i$ and $e_j$, where each are free-text descriptions. In practice, one of these comes from the LM and one comes from the ClinGen VCI/GCI (it can be either because we permute their order for robustness). Using these explanations, we then define the output score as $s[e_i, e_j] \in \{0, 1\}$, where 0 is a non-match and 1 is a match. Also define the context $c$ as potentially being the information about the input sample that generated $e_i$ and $e_j$. Then, the LM judge function **LJ** can be defined as:

$$\mathbf{LJ}(e_i, e_j | c) = s[e_i, e_j], s[e_i, e_j] \in \{0, 1\} \tag{4}$$

We then can sum over the number of matches over a set of $\{(e_i, e_j), ...\}$ pairs to arrive at the overall judge agreement for a set of inputs.

## H.2   Judge parameters

We use SOTA methods for prompting LM judges to avoid bias, as stated in Section 4.3.

**Majority vote @20.** We use majority voting for the judge, i.e., we sample the judge at temperature 1.0 for 20 times (with positional bias adjustment, as below) and take the majority vote from the models.

**Avoiding positional bias.** Position bias occurs when a judge tends to prefer a given option based on the ordering of the options. This has been studied in the context of preference judges, i.e., judges that are asked to prefer example A or example B [57, S4]. If explanations are given as "A B", we then take another input "B A" and measure the judge agreement on this. We then add this to the list of outputs and derive the final score via majority voting.

## H.3   Manual review to ensure judge soundness

To ensure that the LM judge method is sound, we performed a manual review of 40 outputs from GPT-4o and Llama-4 queries on the VCI E-Score task. We took only instances that were correct by evidence code matching, and we selected specific explanations from the sampled outputs (i.e., from the pass@k sampling) to score against ground-truth explanations. The ground-truth explanations in this case were ones provided in the VCI by ClinGen.

We used a 1-5 Likert scale ("strongly disagree", "somewhat disagree", "neutral", "somewhat agree", "strongly agree") to score correspondence between the two explanations. We then used a 0-3 scale for confidence in our answers (0 = "uncertain", 1 = "somewhat confident", 2 = "confident", 3 = "very confident"). All examples with a 1-2 correspondence score were set as "disagree" and examples with 4-5 correspondence score were set to "agree". "Neutral" scores, i.e., scores of 3, were omitted in the evaluation. In addition, we omitted any examples wtih 0 certainty, but all scored examples had at least 1 certainty. These outputs were reviewed by a medical trainee to ensure that matches were scored by a domain expert. We do not provide these manual reviews in our dataset or codebase, but these are available upon request.

# Appendix I   LM judge prompts and selection mechanism

## I.1   LM judge prompts

We test three prompts as given below, 1) the general judge prompt, 2) the task-aware judge prompt, and 3) the evidence-aware judge prompt. The general judge prompt only prompts the model to consider if the two explanations are similar to each other, without giving any awareness of the task. The task-aware prompt gives the model the task that the explanations were generated under as well as the variant, disease, and mode of inheritance. Finally, the evidence-aware prompt gives the judge all the information that the LM got at input to generate the explanation, including the full-text of the paper used to consider the explanation.

You are an impartial judge who is asked to rigorously determine if two explanations refer to the same piece of evidence. The two explanations are considered equivalent if they refer to the same piece of evidence in a scientific paper and make similar conclusions about the findings in the paper. Focus on the content of the two explanations rather than the wording; wording of one explanation may be much more terse than the other, but both could be making similar arguments or referring to the same piece of evidence. It is fine if one explanation gives more detail than the other or includes more information about the experiment, but the explanations are explicitly not corresponding if they contradict each other, refer to different experiments entirely, or come to different conclusions about the same experiment.

Avoid any position biases and ensure that the order in which the responses were presented does not influence your decision. Think step by step when making your judgement, and use logical and unbiased reasoning to deduce your final answer. Provide an explanation for why you made your judgement. Your responses should be given in the following format:

Decision: 'Yes' or 'No'

Explanation: [Your explanation for why you made your judgement]

Provide your answer as 'Yes' or 'No' only. You should answer 'Yes' if the two explanations are equivalent and 'No' if the explanations are not equivalent, according to your instructions.

---

**Task-aware Judge Prompt**

`{general_judge_prompt}`

Here is the task given in this example:

The task for which the explanations were generated is to take a scientific article and determine which evidence code applies to the evidence presented in the paper. Given for this problem is 1) a genetic variant, 2) a disease in which that genetic variant might be pathogenic or benign, 3) a mode of inheritance, and 4) text of a scientific article that is to be analyzed. Here's some information on the evidence codes: Each pathogenic criterion is weighted as very strong (PVS1), strong (PS1-4); moderate (PM1-6), or supporting (PP1-5) and each benign criterion is weighted as stand-alone (BA1), strong (BS1-4) or supporting (BP1-6). The numbering within each category does not convey any differences of weight and are merely labeled to help in referring to the different criteria. For a given variant the user selects the criteria based on the evidence observed for the variant.

Here is the input given for this example:

Variant: `{variant}`

Disease: `{disease}`

Mode of inheritance: `{inheritance}`

Evidence Code: `{evidence_code}`

Description: `{description}`

Note that the "general_judge_prompts" refers to where we insert the "General Judge Prompt" at the beginning of the prompt.

### I.2 Testing prompt types against manually-reviewed outputs

| Prompt Variant | TNR (↑) | TPR(↑) | F1(↑) |
|---|---|---|---|
| Task-agnostic | 0.375±0.003 | **1.000**±0.000 | 0.615±0.003 |
| Task-aware | **0.450**±0.004 | **1.000**±0.000 | **0.744**±0.002 |
| Evidence-aware | 0.375±0.004 | 0.850±0.003 | 0.723±0.002 |

Table 13: Results of judge correspondence to manually-reviewed judgements.

We take the manually-reviewed responses from Appendix H.3 and use these to measure the performance of each prompt type. Table 13 shows the results of this analysis. Task-aware prompting emerges as the top-performing prompting strategy, with evidence-aware prompting following closely behind. This is surprising given that evidence-aware prompting uses the text of the paper being tested to evaluate the correspondence of two explanations. Most likely, the performance is worse for the evidence-aware because it might either 1) introduce bias that is conditional on the paper, such as trying to determine if the two explanations are correct or 2) the full-text of the paper takes up too much of the context window, thereby giving the model less focus on the actual task at hand, i.e., determining if the explanations correspond. Given the superioty of the task-aware judge, we move forward with this prompting strategies for the experiments in the main text.

## Appendix J  LLM usage statement

In the development of CGBENCH, LLMs were used to develop parts of the codebase, refine writing for portions of the manuscript, and to brainstorm certain parts of the methodology. The only portion where LLMs played a significant portion was in writing the scraping code that compiled the benchmark, using iterative development strategies in long multi-turn conversations. However, the majority of the methodology, setup, and experiments were designed by the authors.

## Appendix K    Prompts

### K.1    VCI Evidence Scoring Prompts

This section contains the prompts necessary for querying the VCI Evidence Scoring task.

---

**VCI Evidence Score System Prompt**

You are an evidence critic that is highly skilled in clinical genetics, especially in clinical classification of variants. Your job is to take a PubMed article, which is a scientific, peer reviewed paper, and classify it into one of the "evidence codes" given to you. You will be given a name of a variant of a gene, and you must determine what level of evidence, if any, is provided in the paper. Make your judgement based on the evidence presented in the paper, and use logical reasoning to determine your answer. Think step-by-step, and use your best knowledge of clinical genetics and literature curation for clinical classifications.

Here's some information on the evidence codes: Each pathogenic criterion is weighted as very strong (PVS1), strong (PS1-4); moderate (PM1-6), or supporting (PP1-5) and each benign criterion is weighted as stand-alone (BA1), strong (BS1-4) or supporting (BP1-6). The numbering within each category does not convey any differences of weight and are merely labeled to help in referring to the different criteria. For a given variant the user selects the criteria based on the evidence observed for the variant.

Some codes have may modifiers such as BP1_Strong or PM1_Supporting. These modifiers indicate another granularity of strength of evidence from the core code such as BP1 for BP1_Strong, etc. In the below code specifications, the core code is described with a general description, and finer-grained codes are described with a more detailed description. Consider the finer-grained codes, such as BP1_Strong, as also meeting the criteria of the core code, BP1, unless the more detailed description directly contradicts the core code description. In this case, the finer-grained code description takes precedence.

Here are the evidence codes you must use to classify paper:
`{evidence_code_str}`

Provide your answer in the following format:
Evidence code: <predicted evidence code, such as BP5 or PP4>
Explanation: <Your explanation for why you classified this evidence as this code>

---

The `{evidence_code_str}` is filled by a string of all the evidence codes along with their descriptions. For brevity, we show in the below example only descriptions for two codes:

---

**Example Evidence Code String**

Evidence code: BS2
General code description: Observed in a healthy adult individual for a recessive (homozygous), dominant (heterozygous), or X-linked (hemizygous) disorder, with full penetrance expected at an early age.

Evidence code: BS2_Strong
Detailed code description: Observed in the homozygous state in a healthy adult. Homozygous individual of any age with normal GAA activity.

---

In the above "Example Evidence Code String", the description for both BS2 and BS2_Strong are shown. Since BS2_Strong inherits the description from BS2, we "stack" the descriptions in the prompt, i.e., we put the description for BS2_Strong below BS2 and provide only the additional "detailed code description" which provides specific qualifiers on the base BS2 description. This "stacking" is described in the system prompt for this task. The BS2 and BS2_Strong descriptions are taken from Lysosomal Storage Disorders Variant Curation Expert Panel version 2.0.0.

**ICL prompting.** Below we show two versions of the prompt pieces used to prompt the model for in-context learning (ICL). We try both prompting with only the description of a code and how it is met within a paper and also adding the full-text of the paper. Each prompt piece is for Example $i$, but we concatenate these pieces into the input prompt when using multiple demonstrations.

## K.2    VCI Evidence Verification Prompts

This section contains prompts for the VCI Evidence Verification task.



**VCI Evidence Verification System Prompt**

You are an evidence critic that is highly skilled in clinical genetics, especially in clinical classification of variants. Your job is to take a scientific article and determine if a given evidence code, which is accompanied by a description, is met or not met from the evidence within the paper for a given disease and genetic variant. The code is "met" if the evidence in the paper meets specified rules for the given evidence code. The code is "not met" if, upon evaluation, the evidence in the paper does not meet the criteria specified by the code.

You will be given a variant, disease, mode of inheritance, PubMed paper, and evidence code to accomplish this task. Your determination should be made dependent on the variant, disease, and mode of inheritance given. Make your judgement based on the evidence presented in the paper, and use logical reasoning to determine your answer. Think step-by-step, and use your best knowledge of clinical genetics and literature curation for clinical classifications. Along with your prediction, produce an explanation for which parts of the paper meet the evidence code provided.

Provide your answer in the following format:
Prediction: "met" or "not met"
Explanation: <Your explanation for why you believe this code is met or not met by the evidence in this paper>



Note that the above system prompt does not give the descriptions for all codes; rather, only the code of interest is given to the model in the below template.



**VCI Evidence Verification Input Prompt**

Variant: {variant}
Disease: {disease}
Mode of inheritance: {inheritance}
PubMed ID: {pmid}
PubMed Abstract: {abstract}
PubMed Full Text: {full_text}
Evidence code: {evidence_code}
Description: {description}



The variables here are as described in Section 3.3. The {evidence_code} is the evidence code identifier, such as "PS3" or "BP4" whereas the {description} is the description of the code, like the ones given in the "Example Evidence Code String" in Appendix K.1.

**ICL prompting.** Like for VCI E-Score, we also use both description-only and paper prompts for the in-context demonstrations. Below are shown two prompt pieces that follow these approaches. The only difference between these prompts and the ones for VCI E-Score is the use of the "Prediction" field as well as giving the evidence code and its description. The "description" field and "explanation" field here are distinct: "description" comes from the VCEP description of the evidence code while "explanation" is the ground-truth explanation given by ClinGen in the VCI.



**VCI Evidence Verification: ICL prompt for Example $i$ (description only)**

Variant: {variant}
Disease: {disease}
Mode of inheritance: {inheritance}

Evidence code: {evidence_code}
Description: {description}
Prediction: {met_status}
Explanation: {explanation}



### K.3 GCI Evidence Extraction Prompts

Below, we include the prompts used for GCI evidence extraction tasks. These are similarly broken down into system prompt, input prompt, and in-context (demonstration) prompts.

GCI System Prompt

You are an expert in molecular biology and clinical genetics, especially determining the meaning of experimental evidence and how it contributes the interpretation of a gene-disease relationship. Your job is to extract and score the quality of experimental evidence from a scientific publication and determine it's contribution to the interpretation of a gene-disease relationship. You will be given a gene, disease, mode of inheritance, and paper from PubMed, and you will be asked to provide structured pieces of evidence from this paper that contribute to how the gene may or may not be implicated in the molecular etiology of the disease given.

You must choose from the following list of experimental evidence categories, with defined score ranges and default scores. More instruction on these score ranges and default scores are provided below. Here are the categories:
`{experimental_category_list}`

You will output your response in between tags denoting different portions of the extracted evidence:

<evidence>
<category> The experimental category classifies this extracted piece of evidence as provided by the guidelines. This category describes what type of experimental evidence is described in this case; descriptions of these categories are provided in the list below. Categories must only be chosen from the list above, and the categories you output must exactly match the name of the categories given above. </category>
<explanation> Detailed explanation of the findings for this piece of evidence and how it contributes to the molecular etiology of the disease. </explanation>
<score> Numerical score indicating the strength of the evidence. A higher score denotes stronger evidence while lower denotes weaker evidence. Provide a score within the range provided in increments of 0.25. Your score should start at the default score if the evidence meets the criteria, then you can add or subtract points based on the strength, e.g., you should deduct points if you believe the evidence is weaker than the default guidelines and add points if it is stronger. Scores are defined as above for the given categories. </score>
<score_adjustment_reason> Detailed explanation of why you chose to either deduct or add points to the default score for this evidence. Please provide detailed comments on the strength of the evidence and how this contributes to your assessment. </score_adjustment_reason>
</evidence>

The <evidence></evidence> tags wrap the entire response. You may output multiple groups of evidence tags in your response if the presented paper contains multiple pieces of evidence that meet the criteria.
If you do not use this format, your outputs are invalid. Adhere to the format strictly.
Now extract evidence from the following PubMed article related to the given gene, disease, and mode of inheritance:
Gene: {gene}
Disease: {disease}
Mode of inheritance: {mode_of_inheritance}
PubMed ID: {pmid}
PubMed Abstract: {abstract}
PubMed Full Text: {full_text}
Extracted Evidence: {extracted_evidence}

`experimental_category_list` is a string of the list of options for experimental categories (as defined by the relevant SOP) alongside the range of scores for that category, also given by the SOP. Please see the codebase for the exact specifications of these descriptions as we omit them here for brevity.

**ICL prompting.** For in-context demonstrations, we use the above template wrapped in the `<evidence>` tags. In each, we fill in the variables with the relevant inputs and outputs for the demonstration. These ICL examples are important to inform the LM of the output structure, so we include them on every input prompt for the GCI.

---

**GCI Evidence Extraction: ICL prompt for Example $i$**

Example: {i}
Gene: {gene}
Disease: {disease}
Mode of inheritance: {mode_of_inheritance}
PubMed ID: {pmid}
PubMed Abstract: {abstract}
PubMed Full Text: {full_text}
Extracted Evidence:
<evidence>
<category> {category} </category>
<explanation> {explanation} </explanation>
<score> {score} </score>
<score_adjustment_reason> {score_adjustment_reason} </score_adjustment_reason>
...
</evidence>

---

## Appendix L    Implementation Details

### L.1    Random baseline implementation

**VCI evidence scoring.** We determine the random performance by an instance-based Monte Carlo simulation whereby for each $i$th example, we sample 5 codes (without replacement) 100 times from $\mathcal{C}_i$ evidence codes. At each individual 5 samples, we compute the Precision@5 and Recall@5. We then take the mean performance of these metrics across all samples, and report that mean and standard error as the performance. This is done at the sub-levels for each code.

**VCI Evidence Verification.** The score and standard error for a random baseline on the evidence verification task can be derived analytically. We assume a determination of "met" or "not met" is taken at uniform probability, thus resulting in a Bernoulli distribution with $p = 0.5$. The mean of the distribution is $p$, e.g., 0.5. The standard error of the mean for a Bernoulli distribution is $\sqrt{p(1-p)/n}$. Thus, at 242 evaluation samples with $p = 0.5$, the standard error is 0.032. TPR, TNR, and Positive Rate are all modeled with the same distribution.

**GCI Evidence Extraction.** We perform 1000 rounds of Monte Carlo sampling to get GCI random baseline results. For the evidence category, we uniformly choose 1) $n \sim \{1, 2, 3, 4, 5\}$, 2) $n$ categories, with replacement, from the list of 13 valid evidence categories. For the scoring, we choose a score uniformly in the range given for each of the categories. We then compute metrics as defined in Section 4.2.

### L.2    Scoring details

**Dealing with multiple instances of a category - GCI.** In the case that an LM predicts multiple instances of the same experimental category in the GCI Evidence Extraction task, the multiple predictions are cancelled out in the category score, i.e., we just consider an `any(.)` call across the extracted categories to see if any piece of that category was retrieved by the LM. For score-based evaluation, the matching categories would be scored against all ground-truth entries with the same category.

### L.3 LM parameters

For VCI E-Score and GCI Evidence Extraction, we sample all LMs with 0.5 temperature as we use pass@5 sampling. For VCI E-Ver, we use 0.2 sampling as the temperature. We attempted majority voting for scaling E-Ver but found that this provided no benefit over single-pass sampling; for efficiency, we do not provide full results with majority voting in this appendix. For reasoning models o4-mini and o3-mini, OpenAI does not allow temperature specification. However, for Deepseek-R1, we use the same temperatures as the rest of the models.

We do not modify other default parameters for each of the APIs, such as frequency penalties or maximum token lengths. No inputs in our benchmarked samples required more input tokens than context length bounds, so we did not have to truncate inputs.

### L.4 Obtaining error bars

Since our results are aggregated across many samples, we provide error bars to give a sense of the statistical significance of separation of different metrics.

**VCI Evidence Scoring.** Precision@5 and recall@5 errors are given as standard error when aggregating performance across all evaluation samples.

**VCI Evidence Verification.** Since metrics are calculated across all instances for evidence verification (rather than providing mean results across many samples), we report bootstrapped results for the TPR, TNR, and F1 score. The bootstrapping is done by performing 1000 samples of size N, where N is the number of samples, with replacement and measuring standard error for these 1000 samples. Please consult our codebase for more details.

**GCI Evidence Extraction.** We provide the standard error of the recall@5 and precision@5 after aggregating across all samples in the GCI. For structure adherence, we do not provide error bars. For scoring, our error bars are also given as standard error after aggregating scores across individual samples.

### L.5 Parsing LM outputs

We parse the LM outputs for each task according to the desired output structure. For the VCI task, these desired output structures are simply in the form of "Evidence code:  Explanation: <explanation>" for VCI E-Score and "Prediction: <met or not met> Explanation: <explanation>" for VCI E-Ver. If the parsing functions do not succeed, we consider this as an incorrect prediction by the model for this case. We explicitly study the adherence of LMs to desired output formats for the GCI evidence extraction task, but for VCI E-Score and E-Ver, we do not explicitly mark this. However, we saw that extraction failures occurred very rarely for the VCI tasks during benchmarking, so this is not a focus of our analysis. Please reference our codebase for the implementation of our parsing functions.

### L.6 Computational Requirements for Reproducibility

Our reproduction requires only API keys to OpenAI, Anthropic, and TogetherAI (although the open-source models can be accessed by another cloud provider or by local resources). Thus, only CPUs are needed to reproduce experiments. More information is given in the GitHub README.

## Appendix M  Rebuttals

### M.1 Date cutoff analysis

Next, we perform an analysis of samples published after the cutoff dates of each of the models.

|  | Evidence Scoring | | | | | |
|  | **Primary** | | **Secondary** | | **Tertiary** | |
| **Method** | Precision@5 | Recall@5 | Precision@5 | Recall@5 | Precision@5 | Recall@5 |
|---|---|---|---|---|---|---|
| Random | 0.524±0.219 | 0.980±0.140 | 0.144±0.155 | 0.550±0.497 | 0.038±0.083 | 0.180±0.384 |
| GPT-4o | **0.861**±0.024 | 0.878±0.023 | **0.517**±0.033 | 0.568±0.033 | 0.383±0.032 | 0.427±0.033 |
| GPT-4o (PMID) | 0.841±0.025 | 0.858±0.024 | 0.361±0.029 | 0.505±0.034 | 0.152±0.019 | 0.277±0.030 |
| GPT-4o (Random PMID) | 0.462±0.030 | 0.652±0.033 | 0.161±0.020 | 0.328±0.032 | 0.055±0.011 | 0.125±0.022 |
| LLaMA 4 | 0.837±0.023 | 0.873±0.023 | 0.471±0.032 | 0.532±0.034 | 0.361±0.031 | 0.424±0.034 |
| LLaMA 4 (PMID) | 0.754±0.022 | 0.887±0.022 | 0.343±0.027 | 0.473±0.034 | 0.189±0.022 | 0.294±0.031 |
| LLaMA 4 (Random PMID) | 0.767±0.023 | 0.877±0.023 | 0.327±0.027 | 0.475±0.034 | 0.138±0.018 | 0.260±0.030 |
| o4-mini | 0.743±0.027 | 0.859±0.024 | 0.494±0.032 | 0.600±0.033 | **0.420**±0.032 | 0.495±0.034 |
| o4-mini (PMID) | 0.876±0.022 | 0.912±0.020 | 0.429±0.031 | 0.537±0.034 | 0.238±0.026 | 0.373±0.033 |
| o4-mini (Random PMID) | 0.851±0.022 | 0.917±0.019 | 0.342±0.028 | 0.500±0.034 | 0.196±0.023 | 0.331±0.032 |

Table 14: VCI E-Score. For all metrics, higher is better (↑). **Best**, 2nd-best.

| **Method** | **Hard Negative Errors (↓)** |
|---|---|
| GPT-4o-mini | 0.244±0.057 |
| GPT-4o | 0.244±0.061 |
| Sonnet 3.7 | **0.067**±0.033 |
| Qwen2.5 72B | 0.156±0.048 |
| LLaMA 4 | 0.187±0.046 |
| Deepseek R1 | 0.222±0.056 |
| o3-mini | 0.191±0.054 |
| o4-mini | 0.253±0.060 |

Table 15: Hard negative rates

For consistent comparison, we use the latest knowledge cutoff date of all the models in this group which is Claude Sonnet 3.7 with a cutoff date of November 2024[4]. This allows us to compare all models across the same samples that are after their respective knowledge cutoff dates.

|  | Evidence Scoring | | | | | |
|  | **Primary** | | **Secondary** | | **Tertiary** | |
| **Method** | Precision@5 | Recall@5 | Precision@5 | Recall@5 | Precision@5 | Recall@5 |
|---|---|---|---|---|---|---|
| Random | 0.524±0.219 | 0.980±0.140 | 0.144±0.155 | 0.550±0.497 | 0.038±0.083 | 0.180±0.384 |
| GPT-4o-mini | 0.267±0.114 | 0.267±0.114 | 0.227±0.100 | 0.267±0.114 | 0.227±0.100 | 0.267±0.114 |
| GPT-4o | 0.333±0.122 | 0.333±0.122 | 0.267±0.114 | 0.267±0.114 | 0.267±0.114 | 0.267±0.114 |
| Sonnet 3.7 | 0.373±0.121 | 0.400±0.126 | 0.307±0.114 | 0.333±0.122 | 0.307±0.114 | 0.333±0.122 |
| Qwen2.5 72B | 0.293±0.108 | 0.333±0.122 | 0.227±0.098 | 0.267±0.114 | 0.227±0.098 | 0.267±0.114 |
| LLaMA 4 | 0.333±0.122 | 0.333±0.122 | 0.267±0.114 | 0.267±0.114 | 0.267±0.114 | 0.267±0.114 |
| Deepseek R1 | 0.453±0.126 | 0.467±0.129 | 0.400±0.121 | 0.467±0.129 | 0.400±0.121 | 0.467±0.129 |
| o3-mini | 0.400±0.118 | 0.467±0.129 | 0.387±0.114 | 0.467±0.129 | 0.387±0.114 | 0.467±0.129 |
| o4-mini | 0.387±0.102 | 0.667±0.122 | 0.280±0.090 | 0.600±0.126 | 0.280±0.090 | 0.600±0.126 |

Table 16: VCI E-Score after Dec. 01, 2024 cutoff. For all metrics, higher is better (↑). **Best**, 2nd-best.

---

[4]See https://docs.anthropic.com/en/docs/about-claude/models/overview

| | Evidence Scoring | | | | | |
| | Primary | | Secondary | | Tertiary | |
| **Method** | Precision@5 | Recall@5 | Precision@5 | Recall@5 | Precision@5 | Recall@5 |
|---|---|---|---|---|---|---|
| Random | $0.524_{\pm0.219}$ | $0.980_{\pm0.140}$ | $0.144_{\pm0.155}$ | $0.550_{\pm0.497}$ | $0.038_{\pm0.083}$ | $0.180_{\pm0.384}$ |
| GPT-4o-mini | $0.886_{\pm0.023}$ | $0.895_{\pm0.022}$ | $0.482_{\pm0.034}$ | $0.547_{\pm0.035}$ | $0.282_{\pm0.029}$ | $0.347_{\pm0.033}$ |
| GPT-4o | $0.903_{\pm0.021}$ | $0.921_{\pm0.020}$ | $0.537_{\pm0.034}$ | $0.592_{\pm0.034}$ | $0.393_{\pm0.033}$ | $0.439_{\pm0.035}$ |
| Sonnet 3.7 | $0.864_{\pm0.023}$ | $0.905_{\pm0.021}$ | $0.438_{\pm0.034}$ | $0.495_{\pm0.035}$ | $0.313_{\pm0.031}$ | $0.361_{\pm0.033}$ |
| Qwen2.5 72B | $0.847_{\pm0.022}$ | $0.905_{\pm0.021}$ | $0.501_{\pm0.032}$ | $0.582_{\pm0.035}$ | $0.274_{\pm0.029}$ | $0.326_{\pm0.033}$ |
| LLaMA 4 | $0.877_{\pm0.021}$ | $0.916_{\pm0.020}$ | $0.487_{\pm0.033}$ | $0.553_{\pm0.035}$ | $0.368_{\pm0.032}$ | $0.437_{\pm0.035}$ |
| Deepseek R1 | $0.805_{\pm0.022}$ | $0.932_{\pm0.018}$ | $0.492_{\pm0.031}$ | $0.642_{\pm0.034}$ | $0.419_{\pm0.032}$ | $0.521_{\pm0.035}$ |
| o3-mini | $0.797_{\pm0.026}$ | $0.874_{\pm0.024}$ | $0.489_{\pm0.034}$ | $0.561_{\pm0.035}$ | $0.403_{\pm0.033}$ | $0.466_{\pm0.035}$ |
| o4-mini | $0.772_{\pm0.027}$ | $0.874_{\pm0.024}$ | $0.511_{\pm0.033}$ | $0.600_{\pm0.034}$ | $0.431_{\pm0.034}$ | $0.487_{\pm0.035}$ |

Table 17: VCI E-Score. For all metrics, higher is better (↑). **Best**, 2nd-best.

# Appendix References

[S1] Antonio Jimeno Yepes and Karin Verspoor. Literature mining of genetic variants for curation: quantifying the importance of supplementary material. *Database*, 2014:bau003, 2014.

[S2] Emilie Pasche, Anaïs Mottaz, Julien Gobeill, Pierre-André Michel, Déborah Caucheteur, Nona Naderi, and Patrick Ruch. Assessing the use of supplementary materials to improve genomic variant discovery. *Database*, 2023:baad017, 2023.

[S3] Alexis Allot, Chih-Hsuan Wei, Lon Phan, Timothy Hefferon, Melissa Landrum, Heidi L Rehm, and Zhiyong Lu. Tracking genetic variants in the biomedical literature using litvar 2.0. *Nature genetics*, 55(6):901–903, 2023.

[S4] Owen Queen, Yepeng Huang, Robert Calef, Valentina Giunchiglia, Tianlong Chen, George Dasoulas, LeAnn Tai, Yasha Ektefaie, Ayush Noori, Joseph Brown, et al. Procyon: A multimodal foundation model for protein phenotypes. *BioRxiv*, pages 2024–12, 2024.

[S5] Allen Nie, Arturo L Pineda, Matt W Wright, Hannah Wand, Bryan Wulf, Helio A Costa, Ronak Y Patel, Carlos D Bustamante, and James Zou. Litgen: Genetic literature recommendation guided by human explanations. In *PACIFIC SYMPOSIUM ON BIOCOMPUTING 2020*, pages 67–78. World Scientific, 2019.

[S6] Jun Cheng, Guido Novati, Joshua Pan, Clare Bycroft, Akvilė Žemgulytė, Taylor Applebaum, Alexander Pritzel, Lai Hong Wong, Michal Zielinski, Tobias Sargeant, et al. Accurate proteome-wide missense variant effect prediction with alphamissense. *Science*, 381(6664):eadg7492, 2023.

[S7] Garyk Brixi, Matthew G Durrant, Jerome Ku, Michael Poli, Greg Brockman, Daniel Chang, Gabriel A Gonzalez, Samuel H King, David B Li, Aditi T Merchant, et al. Genome modeling and design across all domains of life with evo 2. *BioRxiv*, pages 2025–02, 2025.

[S8] Nilah M Ioannidis, Joseph H Rothstein, Vikas Pejaver, Sumit Middha, Shannon K McDonnell, Saurabh Baheti, Anthony Musolf, Qing Li, Emily Holzinger, Danielle Karyadi, et al. Revel: an ensemble method for predicting the pathogenicity of rare missense variants. *The American Journal of Human Genetics*, 99(4):877–885, 2016.

[S9] Jonathan Frazer, Pascal Notin, Mafalda Dias, Aidan Gomez, Joseph K Min, Kelly Brock, Yarin Gal, and Debora S Marks. Disease variant prediction with deep generative models of evolutionary data. *Nature*, 599(7883):91–95, 2021.

[S10] Joshua Meier, Roshan Rao, Robert Verkuil, Jason Liu, Tom Sercu, and Alex Rives. Language models enable zero-shot prediction of the effects of mutations on protein function. *Advances in neural information processing systems*, 34:29287–29303, 2021.

[S11] Weijiang Li, Xiaomin Li, Ethan Lavallee, Alice Saparov, Marinka Zitnik, and Christopher Cassa. From text to translation: Using language models to prioritize variants for clinical review. *medRxiv*, 2024.

[S12] Yang Li, Zihou Guo, Keqi Wang, Xin Gao, and Guohua Wang. End-to-end interpretable disease–gene association prediction. *Briefings in bioinformatics*, 24(3):bbad118, 2023.

[S13] Emily Alsentzer, Michelle M Li, Shilpa N Kobren, Ayush Noori, Undiagnosed Diseases Network, Isaac S Kohane, and Marinka Zitnik. Few shot learning for phenotype-driven diagnosis of patients with rare genetic diseases. *medRxiv*, 2024.

[S14] Zehui Li, Vallijah Subasri, Guy-Bart Stan, Yiren Zhao, and Bo Wang. Gv-rep: A large-scale dataset for genetic variant representation learning. In *The Thirty-eight Conference on Neural Information Processing Systems Datasets and Benchmarks Track*, 2024.

[S15] National Cancer Institute. pathogenic variant. https://www.cancer.gov/publications/dictionaries/genetics-dictionary/def/pathogenic-variant. Accessed: 2025-05-12.

[S16] Roberta A. Pagon, Margaret P. Adam, Holly H. Ardinger, Stephanie E. Wallace, Akiko Amemiya, Lisa J. H. Bean, and Kate Stephens. Benign variant. *GeneReviews® Glossary*. https://www.ncbi.nlm.nih.gov/books/NBK5191/def-item/benign-variant/, 2017. Seattle (WA): University of Washington, Seattle. Accessed 11 May 2025.

[S17] Timothy H Ciesielski, Giorgio Sirugo, Sudha K Iyengar, and Scott M Williams. Characterizing the pathogenicity of genetic variants: the consequences of context. *NPJ Genomic Medicine*, 9(1):3, 2024.

[S18] Sue Richards, Nazneen Aziz, Sherri Bale, David Bick, Soma Das, Julie Gastier-Foster, Wayne W Grody, Madhuri Hegde, Elaine Lyon, Elaine Spector, et al. Standards and guidelines for the interpretation of sequence variants: a joint consensus recommendation of the american college of medical genetics and genomics and the association for molecular pathology. *Genetics in medicine*, 17(5):405–423, 2015.

[S19] The Clinical Genome Resource Gene Curation Working Group. Gene-disease validity curation process. https://www.clinicalgenome.org/site/assets/files/9851/gene-disease_validity_standard_operating_procedures-_version_11_docx.pdf, 2024. Accessed: 2025-05-19.

[S20] Michael D Skarlinski, Sam Cox, Jon M Laurent, James D Braza, Michaela Hinks, Michael J Hammerling, Manvitha Ponnapati, Samuel G Rodriques, and Andrew D White. Language agents achieve superhuman synthesis of scientific knowledge. *arXiv preprint arXiv:2409.13740*, 2024.

[S21] David Rein, Betty Li Hou, Asa Cooper Stickland, Jackson Petty, Richard Yuanzhe Pang, Julien Dirani, Julian Michael, and Samuel R Bowman. Gpqa: A graduate-level google-proof q&a benchmark. In *First Conference on Language Modeling*, 2024.

[S22] Peiyi Wang, Lei Li, Liang Chen, Zefan Cai, Dawei Zhu, Binghuai Lin, Yunbo Cao, Qi Liu, Tianyu Liu, and Zhifang Sui. Large language models are not fair evaluators. *arXiv preprint arXiv:2305.17926*, 2023.

