# OpenReview forum: "CGBench: Benchmarking Language Model Scientific Reasoning for Clinical Genetics Research"
_NeurIPS.cc/2025/Datasets_and_Benchmarks_Track — NeurIPS 2025 Datasets and Benchmarks Track poster_

### Official Review · Reviewer_4ngH · 2025-06-21

**Rating:** 4
**Confidence:** 3

**Summary:**

In this benchmark paper, the authors proposed a ClinGen database that tests the complex LLM reasoning on scientific publications regarding genetic causes associated with human disease. In particular, the CGBENCH benchmark is established on high-quality ClinGen’s Evidence Repository (ERepo) annotated by professional clinical experts.  For this difficult and clinically relevant benchmark, the authors demonstrate the drawbacks of existing LLMs and spur the novel research directions for future real-world, translational scientific uses.

**Dataset Code Accessibility:**

Yes

**Dataset Code Comments:**

The Huggingface project page for this CGBench and the code base are available.

**Ethical Considerations:**

No, there are no or only very minor ethics concerns

**Final Justification:**

Thank you for the authors' feedback, and I keep my rating.

**Limitations Weaknesses:**

I think one aspect (which may be hard to achieve?) that can be improved is to include available genomic data for each supporting study. Currently, the evidence seems solely text-based, without the support of raw genomic data. This aspect can greatly increase the potential impact brought by the benchmark.

Accordingly, discussing the plausible pathways towards multi-modal LLMs development can be very interesting to add to the study.

**Strengths Contributions:**

1. This paper is well-motivated. Translating genomic data into clinical diagnostics and treatment is a legitimate task for real-world practice.

2. The benchmark is built on high-quality resources, the Clinical Genomic Resource (ClinGen). which is very valuable given its collection of reputable biomedical evidence from a large number of experts.

3. The baseline experiments are validated on 8 widely-used LLMs, providing a good starting point for progressing this research direction.

---

> ### Author Response · Authors · 2025-07-31
> **Response to Reviewer Feedback**
>
> Thank you for your comments on our work! We agree that CGBench reflects real-world, impactful applications in biomedicine, where benchmarking SOTA LMs on human-generated, high-quality data are of the upmost importance. In the following, we will address each of your comments from the “Limitations Weaknesses” (LW) section and the “Additional Feedback” (AF) section, broken down by the order of comments.
>
> LW1: Thank you for this interesting suggestion on the use of multimodal data in our benchmark. We would like to point out an important nuance here: while our work does focus on genetics tasks, we are primarily interested in the use of LMs to aide in interpretation of scientific publications centered around gene-disease validity and variant annotation. As we point out in Related Work and Appendix C, this task is distinctly different from variant effect prediction or genomics-centric tasks, which are concerned with interpreting the phenotypes of genes and variants from alterations in DNA sequence. We operate at another endpoint of variant and gene interpretation: we seek to understand how LMs can take already-published works to judge and categorize scientific evidence for downstream clinical application. Integrating multimodal data from genomics into LMs is an important research area; however, CGBench operates in a related but different regime, focusing on interpretation of scientific publications.

---

### Official Review · Reviewer_nJj5 · 2025-07-02

**Rating:** 4
**Confidence:** 3

**Summary:**

The paper introduces CGBENCH, a new benchmark for evaluating the scientific reasoning capabilities of large language models (LLMs) in the field of clinical genetics. CGBENCH is built using the expert-curated ClinGen database and presents LLMs with complex reasoning tasks based on scientific publications. The benchmark consists of three main tasks: (1) Extract experimental results according to precise guidelines; (2) Judge the strength of evidence presented in scientific papers; (3) Categorize and describe assays and experiments relevant to specific genes and variants. The study tests eight different LLMs and finds that while they show potential, significant gaps remain, particularly in following fine-grained instructions. The results indicate that reasoning models perform better on detailed extraction tasks, whereas non-reasoning models are more adept at higher-level questions. The paper also highlights that in-context learning can significantly improve performance, but models still frequently "hallucinate" or misinterpret results even when they appear to classify evidence correctly.

**Additional Feedback:**

Minor: the presentation of this paper could be further improved. Too much /vspace usage.

**Dataset Code Accessibility:**

Yes

**Dataset Code Comments:**

The code and data are well-organized and easy to follow/reproduce.

**Ethical Considerations:**

No, there are no or only very minor ethics concerns

**Final Justification:**

Thank you for the response. All my previous concerns have been solved point-by-point and I would like to keep my overall positive score.

**Limitations Weaknesses:**

-  The results for ICL are inconsistent. For the VCI Evidence Scoring task, performance generally increases with more examples. However, for the VCI Evidence Verification task, ICL provides minimal benefit and sometimes even degrades performance. This inconsistency suggests that the effectiveness of the ICL strategy is highly task-dependent, which not sure if it is because of task itself or the potential bias in data collection.
- The LLM-as-a-judge evaluation shows that even when a model correctly classifies evidence, its explanation may not match the expert's. A mismatch in explanations doesn't automatically equate to a hallucination as suggested by the paper. It is possible for two actors (human or AI) to arrive at the same correct conclusion by interpreting the same evidence through slightly different, yet still valid, reasoning paths. The judging methodology focuses on whether the explanations refer to the "same piece of evidence" and "make the same conclusions," which may penalize valid alternative interpretations.
- The paper notes that Deepseek R1, a "reasoning" model, fails on this structural requirement nearly 40% of the time. This suggests that the benchmark may be conflating a model's scientific reasoning capability with its ability to follow formatting instructions. A model might correctly identify the evidence but fail the evaluation because its output cannot be parsed, thus underestimating its true reasoning performance. Would this be "unfair" for reasoning model?

**Strengths Contributions:**

- Unlike benchmarks that use simplified or multiple-choice questions, CGBENCH uses tasks that mirror the complex, multi-step evidence synthesis required by expert human curators in real-world clinical genetics research.
- The benchmark is sourced from the ClinGen Evidence Repository, ensuring that the tasks and ground-truth data are of high quality and based on expert consensus.
- CGBENCH's tasks require LLMs to understand and apply detailed and specific guidelines (VCEP and SOP protocols) that change depending on the context of the disease, gene, and variant.  This pushes the evaluation beyond simple information retrieval to assess deeper reasoning.
- The work introduces a systematic method for evaluating the free-text explanations generated by LMs against those provided by expert curators, using an LM-as-a-judge approach.

---

> ### Author Response · Authors · 2025-07-31
> **Response to Reviewer Feedback**
>
> Thank you for your insightful comments on our manuscript! In the following, we will address each of your comments from the “Limitations Weaknesses” (LW) section and the “Additional Feedback” (AF) section, broken down by the order of comments.
>
> LW1: Thank you for identifying this peculiarity in our results. It is indeed an open question as to the extent that in-context examples affect the performance of LMs for these tasks. This is one of the many areas in which CGBench can be explored as a benchmark, and we do not claim that our choice of in-context examples here is optimal.
>
> As an exploration into this, we ran multiple sets of in-context examples when building prompts for the VCI E-Score and E-Ver tasks. In the paper, we use one static choice of examples for consistency, but in this case, we vary the examples chosen. Practically, we then obtain 5 different combinations of examples for the in-context prompting and show the results in the below tables for both E-Score and E-Ver for GPT-4o. Error bars are omitted for brevity, and for E-Score, we only show the Tertiary code prediction. S* refers to the seed and number for each trial.
>
> Evidence scoring:
>
> | Method | Precision@5 | Recall@5 |
> |-------|-------------:|---------:|
> | S1    | 0.482        | 0.488    |
> | S2    | 0.492        | 0.500    |
> | S3    | 0.482        | 0.527    |
> | S4    | 0.494        | 0.505    |
> | S5    | 0.473        | 0.512    |
>
> Evidence Verification:
>
> | Seed | F1@1 |
> |------|-------------------------------:|
> | S0   | 0.657                          |
> | S1   | 0.655                          |
> | S2   | 0.651                          |
> | S3   | 0.652                          |
> | S4   | 0.658                          |
>
> Across both tasks, choice of in-context examples for prompting results in remarkably stable performance, with the range of precision for VCI E-Score being only 0.021 and the range for VCI E-Suff being only 0.007. We did not repeat these experiments across all shot-levels due to costs, but this illustrates why simple prompting approaches such as in-context learning might not be sufficient for CGBench, solidifying it as a challenging benchmark.
>
> LW2 ("The LLM-as-a-judge evaluation..."): Thank you for your insightful comment on our work and for pointing out the specific details of how we refer to this result.
>
> We agree that it is possible that the LMs could possibly identify a correct piece of evidence that does not match the human explanation in the paper. In this case, it would be a "missed" false positive, i.e., a case in which the LM found a code that was satisfied in the paper but not identified by a human curator, or an explanation for the code that did not match the human explanation. We believe that these missed positives do not occur often in practice for two main reasons. First, papers examined by curators are often analyzed holistically to determine their role in curations. Second, the codes and categories are defined in an orthogonal manner, meaning that codes are not contained, by definition, in the same paper. The GCI evidence codes can coexist more commonly than VCI codes, which is seen in the data where 36% of papers are cited in multiple evidence extractions across the GCI samples.
>
> We believe the LM judge approach for analyzing similar explanations is a useful framework for understand if models are extracting the same evidence as given by the human curators. However, we agree that our strong language about hallucinations might be misleading, so we will change this language in the revised draft of the paper.
>
> LW3 ("The paper notes..."): Thank you for noting this result as it points to an important distinction in how we treat structured output. You are specifically referring to the result in the GCI experimental evidence extraction task, where Deepseek only succeeds in meeting the structured output parser in 61.61% of the cases (Table 4). The requested structured output in this case is quite complex, involving a list of JSON objects containing each field for the evidence extraction (see Appendix K.3 for more details). Thus, we choose not to penalize the models for not correctly extracting the evidence in the proper format, meaning that we only score Deepseek on the 61.61% of samples that pass the criteria.
>
> The question of structured output adherence is indeed an important one, especially for scientific research and automated extraction tasks. For tasks such as these, with rigorous guidelines and formats that have been developed by experts, LLMs are expected to meet the format to comply with the guidelines. Therefore, it is useful to note when some models, such as Deepseek R1, do not meet these requirements. We hope this opens up discussion to the community about the use of structured output in AI for biomedical and scientific research going forward.
>
> ("Minor: the presentation..."): Thank you for this feedback, we will make sure to decrease our use of vspace in the revised manuscript.

---

> > ### Comment · Reviewer_nJj5 · 2025-08-01
> >
> > Thank you for the response. All my previous concerns have been solved point-by-point and I would like to keep my overall positive score.

---

> > > ### Author Response · Authors · 2025-08-01
> > >
> > > Thank you for your kind words about our response! Please let us know if you have any other concerns or if there are any other points we can address to improve our work.

---

### Official Review · Reviewer_rfNg · 2025-07-03

**Rating:** 5
**Confidence:** 4

**Summary:**

CGBENCH converts thousands of expert-curated ClinGen records into a three-part benchmark that mirrors real curation work: (1) picking the correct ACMG/AMP evidence codes for variant-disease papers, (2) deciding if a proposed code is truly “met,” and (3) extracting structured experimental-evidence tuples from gene-disease studies. Covering 191 variants, 1 291 genes and 860 diseases (2 680 individual curator explanations), it stresses fine-grained protocol compliance rather than shallow biomedical recall. Eight leading LLMs (closed and open, reasoning-tuned and standard) were tested zero-shot and with up to 30 demonstrations: they handled coarse pathogenic-vs-benign judgments but their precision plunged on tertiary codes (≈ 0.42), evidence-verification F1 peaked at 0.63, and the best model captured only ~42 % of experimental evidence items with frequent strength-direction mistakes. These gaps, plus widespread rationale hallucinations, highlight the need for retrieval, tool use and alignment methods tailored to the exacting workflows of clinical-genetics research.

**Additional Feedback:**

• Could you quantify uncertainty at the level of each VCEP and share per-panel counts, then describe any plan to enlarge panels with very few samples, for example by adding new curations or merging related panels.

• How will you show that proprietary models did not see ClinGen text during pretraining; a practical step would be to publish a hash-based overlap audit and to report model accuracy on papers added after the main model cut-off dates.

• Please give a fuller validation of the LM judge by stating the size of the manual review set, the qualifications of human raters, the resulting inter annotator agreement, and, if possible, by releasing that annotated subset so other judging schemes can be tested.

**Dataset Code Accessibility:**

Yes

**Ethical Considerations:**

No, there are no or only very minor ethics concerns

**Final Justification:**

The rebuttal has successfully addressed my concerns, and thus I decided to raise the score.

**Limitations Weaknesses:**

• Domain is limited to clinical genetics, so findings may not generalise to other biomedical areas.

• The benchmark pools 205 variant-paper cases across 33 VCEPs, which works out to roughly 7 curated examples per panel on average. Such a thin sample makes any panel-level performance comparison statistically fragile: confidence intervals would be wide and overfitting to idiosyncratic cases becomes likely.

• Nowhere in the methods, results, or stated limitations does the paper examine whether proprietary models (e.g., GPT-4o, Claude Sonnet) might already contain ClinGen curation text or even the exact curator explanations used as ground truth. The limitations note that further work is needed but does not broach this issue. Without a de-duplication audit, “zero-shot” scores may be artificially inflated.

• The authors calibrate their automatic judge against “a small-scale manual review” and report 0.744 F1 for the best prompting variant, but they do not specify the sample size or inter-annotator agreement. A tiny calibration set means the true error rate of the judge—and thus of all explanation-quality numbers—remains uncertain.

**Strengths Contributions:**

• Clinically grounded dataset: built from thousands of expert curated ClinGen records that reflect real variant, gene, and disease evidence.

• Task suite matches actual curation workflow, covering evidence coding, code verification, and extraction of experimental evidence from the literature.

• Evaluation includes curator rationales and an LM as judge protocol, so researchers can measure both answer correctness and explanation quality.

• Baseline study spans eight state of the art language models in zero shot and few shot settings, clearly quantifying where current systems fall short.

---

> ### Author Response · Authors · 2025-07-31
> **Response to Reviewer Feedback - Part 1**
>
> Thank you for your detailed and thorough review of our work!
> In the following, we will address each of your comments from the “Limitations Weaknesses” (LW) section and the “Additional Feedback” (AF) section, broken down by the order of comments.
>
> LW1 ("Domain is limited..."): Thank you for identifying this limitation of our work. We agree that CGBench is focused on clinical genetics, but the process of variant and gene curation draws on many areas of biomedical research. Both variant and gene curation involve interpretation of results from association studies, functional genomics, case-level clinical data, in-silico predictors, therapeutic evidence, or database mining, just to name a few. The experimental approaches that must be interpreted range from CRISPR knockout studies to rescue experiments to large-cohort scRNA-seq or genomic sequencing readouts. Clinical genetics is a translational endpoint of biomedical research, so we argue that this allows us to understand LM capabilities in broader biomedical research. Please see Richards et al. [1] and Wright et al. [2] for more information on the type of evidence used in ClinGen.
>
> LW2 ("The benchmark pools..."): Thank you for your important comment on the variant curation expert panels (VCEPs) that serve as categorizations for the samples in the VCI tasks.
> While we treat the VCEP guidelines as separate entities, these guidelines are in practice very similar.
> The ACMG/AMP protocol (Richards et al., [1]) serves as the basis for how a VCEP determines their guidelines, and often only a few codes are changed within the base guidelines provided by Richards et al.
>
> This is evidenced by the high similarity in guidelines across the different VCEPs.
> Defining similar VCEP guidelines as those with a ROUGE-L score >0.95, we find that 62\% of the codes have over 75\% reuse across all VCEP guidelines, meaning that 75\% of VCEPs that use such a code decide to reuse a standard definition.
> This high similarity across VCEPs illustrates the similarity of the variant curation task, allowing us to draw effective conclusions even given a small number of VCEPs for some samples (see comment in AF1).
>
> LW3 ("Nowhere in the methods..."): Thank you for identifying this important point about test set contamination. This is a problem broadly with benchmarking internet-scale LMs, especially those that do not release training set details that can be used for auditing the contamination probability.
>
> CGBench is designed to minimize contamination of the samples in three main ways.
> First, the ClinGen database's structure does not inject the full-text of the PubMed article beside the evidence classification for each sample. Only the PubMed ID is shown as a reference. Please see the ClinGen website (clinicalgenome.org) for more information on this.
> Second, no definition of the evidence code is shown on the ClinGen webpages. These evidence codes are defined in a separate database within the ClinGen site, so it is unlikely that LMs would associate these samples during large-scale pretraining.
> Third, we design custom prompting for each example in the ClinGen database; since our prompts are custom-designed for the tasks we seek to benchmark, we are confident the models have not seen such prompts at training time.
>
> We test this hypothesis by replace the full-text of the paper with the PMID in the original prompt. Due to character constraints, we include only the VCI E-score task in this experiment.
> This tests if the language model is utilizing the paper's information to make it's determination. In another experiment (Random PMID), we replace the PMID with a random PMID, observing if the model has memorized the PMID to relate the variant/disease.
>
> |Method|Primary P@5|Primary R@5|Secondary P@5|Secondary R@5|Tertiary P@5|Tertiary R@5|
> |---|---:|---:|---:|---:|---:|---:|
> |GPT-4o (PMID)|0.841|0.858|0.361|0.505|0.152|0.277|
> |GPT-4o (Random PMID)|0.462|0.652|0.161|0.328|0.055|0.125|
> |LLaMA 4 (PMID)|0.754|0.887|0.343|0.473|0.189|0.294|
> |LLaMA 4 (Random PMID)|0.767|0.877|0.327|0.475|0.138|0.260|
> |o4-mini (PMID)|0.876|0.912|0.429|0.537|0.238|0.373|
> |o4-mini (Random PMID)|0.851|0.917|0.342|0.500|0.196|0.331|
>
> The performance drop for tertiary code prediction suggests that the model is relying on the content of the papers to make its determination about the code classification.
> Second, the drop is less evident for Primary and Secondary codes. This is potentially because the model tries to assign pathogenic or benign labels to the variants without considering the evidence.
> The model may have prior knowledge about the variant's classification obtained during broad training on the literature, and it is thus answering based on the known classification of the variant.
> These results tell us that primary and secondary code tasks may contain some correlates that impact LM classification, but tertiary code classification requires comprehension of the papers given to the model.

---

> > ### Author Response · Authors · 2025-07-31
> > **Response to Reviewer Feedback - Part 2**
> >
> > LW4 ("The authors calibrate..."): Thank you for pointing out this detail about our LM judge approach. In Appendix H.3, we state that "To ensure that the LM judge method is sound, we performed a manual review of 40 outputs from GPT-4o and Llama-4 queries on the VCI E-Score task." However, we do not provide inter-annotator agreement statistics for the set as annotators reviewed separate samples to maximize sample size.
> >
> > We agree that the relatively small annotation size is not ideal for full analysis of LM explanations. Ideally, we would gather a large multi-participant cohort of experts to derive judgements and then fit a judge approach onto those, but this is out of scope for this work as validation of LM judge approaches is still an active field of study [3]. However, our included 40 samples allow us enough room to approximate the error rate of the judge in order to select the best prompting strategy, which is evidenced by the size of the error bars on our (all below 0.003, Table 9 in Appendix).
> >
> > AF1 ("Could you quantify..."): Thank you for this request to improve our work and presentation. We can indeed add per-VCEP performance in our revised manuscript as well as per-VCEP counts.
> > Unfortunately due to space constraints, we cannot provide such statistics in the rebuttal, but we can comment on VCEP sample counts.
> > In the E-Score task, samples per VCEP range from 1 to 41 with a median of 5.
> > In the E-Ver task, the range is from 1 to 68 with a median of 5.
> > VCEPs with large numbers of samples are often those with well-studied genetic basis for disease, such as Phenylketonuria and RASopathy.
> > Several VCEPs have sample sizes of only 1; however, this does not invalidate such samples as the transfer across VCEPs is reasonable given the similarity in guidelines.
> >
> > We do not plan to enlarge panels at this time, but we do note that ClinGen is an ever-growing resource, so it is possible for our benchmark to be updated with new curations as the database expands.
> >
> > AF2 ("How will you..."): Thank you for your suggestion on ensuring minimal data contamination in CGBench. As stated in response to LW3, we believe that CGBench's construction minimizes the potential for contamination, but we cannot guarantee this, especially for proprietary models. Data contamination is an active research area, and no method can give a certain guarantee whether our data is contained in a pretraining corpus. Our experiments in response to LW3 show that there is minimal memorization of the samples given in CGBench.
> >
> > We examine performance on samples published after the model cutoff dates. We use the latest knowledge cutoff date of all the models, which is Claude Sonnet 3.7 with a cutoff date of November 2024 see {https://docs.anthropic.com/en/docs/about-claude/models/overview}.
> > This allows us to compare all models across the same samples that are after their respective knowledge cutoff dates. We split results into those before the cutoff date (B) and after the cutoff date (A).
> >
> > |Method|Precision@5|Recall@5|
> > |---|---:|---:|
> > |Random (baseline)|0.038±0.083|0.180±0.384|
> > |GPT-4o (B)|0.393±0.033|0.439±0.035|
> > |GPT-4o (A)|0.267±0.114|0.267±0.114|
> > |LLaMA 4 (B)|0.368±0.032|0.437±0.035|
> > |LLaMA 4 (A)|0.267±0.114|0.267±0.114|
> > |o4-mini (B)|0.431±0.034|0.487±0.035|
> > |o4-mini (A)|0.280±0.090|0.600±0.126|
> >
> > These results show a drop in performance across all models from before the cutoff to after the cutoff, but this drop is modest.
> > Such date cutoff ability is a benefit of CGBench, allowing users to audit the effect of timescale for when a curation was published against LM performance.
> >
> > AF3 ("Please give a..."): Thank you for this important suggestion to improve our manuscript. We note that in our original draft, we do reference much of these details in Appendix H.3.
> >
> > The qualifications of the human raters are discussed in our manuscript in Section 4.3: "To calibrate the LM judge approach, we perform a manual review of a subset of LM and ground-truth explanations; review was assisted by a medical trainee with experience in clinical genetics research." The other annotator who participated in the study was a graduate student with a background in biomedical AI research.
> >
> > Following your suggestion, we will release the annotated subset of judgments, including the original LLM explanation as well as the ground-truth explanation and reference to the sample in our dataset. These outputs can be used to test other LLM-as-a-judge approaches, as you mentioned.
> >
> > [1] Richards et al., "Standards and guidelines for the interpretation of sequence variants: a joint consensus recommendation of the American College of Medical Genetics and Genomics and the Association for Molecular Pathology", Genetics in Medicine 2015
> >
> > [2] Wright et al., "Generating clinical-grade gene–disease validity classifications through the ClinGen data platforms", Annual Review of Biomedical Data Science 2024
> >
> > [3] Guerdan et al., "Validating LLM-as-a-judge Systems in the Absence of Gold Labels", ICML 2025

---

> > > ### Comment · Reviewer_rfNg · 2025-08-07
> > > **Concerns addressed**
> > >
> > > Thank you for answering my questions and addressing my concerns. The rebuttal is convincing and thus I have decided to raise the score.

---

> > > > ### Author Response · Authors · 2025-08-07
> > > >
> > > > Thank you for taking the time to consider our rebuttal and raise your score!

---

### Official Review · Reviewer_GmP3 · 2025-07-07

**Rating:** 5
**Confidence:** 4

**Summary:**

The paper introduces a new benchmark for evaluating the ability of LLMs to perform reasoning in clinical genetics. The paper develops 3 tasks for benchmarking, including evidence scoring and verification for variant curation and evidence extraction for gene curation. The results of the paper show that 8 tested LLM models struggle with these tasks, often hallucinating or misinterpreting results

**Additional Feedback:**

While the abstract length might be within the requirements (2000 charasters), it reads too long and can be shortened by 1/3 at least.

**Dataset Code Accessibility:**

Yes

**Dataset Code Comments:**

The code is on Github and Dataset on Hugging Face.

**Ethical Considerations:**

No, there are no or only very minor ethics concerns

**Final Justification:**

The authors have addressed my questions quring rebuttal.

**Limitations Weaknesses:**

$C_{vcep}$ and $\hat{e}_k^y$ are not properly defined for eq1.

In Table 1, why recall@5 for random model is 0.98? It is surprisingly high, especially while compared to that column results.

The two VCI tasks have smaller sample size.

**Strengths Contributions:**

The paper has an interesting setup. The benchmarking tasks are well-defined and seem realistic since they follow real-world scientific protocols used in clinical genetics. As a result, there is an access to an expert-curated ground truth that the paper is using for benchmarking.

The manuscrip introbuces 3 tasks (multiclass, binary classification, and prediction of structured set of properties). Each of them has set of reasonable metrics and is tested on 8 LLM models, allowing for good results analysis.

The explanation evaluation is performed on different levels of context information, which makes evaluation more nuanced. Task-aware methods performs the best, which makes sense.

The paper tests at different ICL shots, which shows non-linear trends and gives interesting direction for future discussion.

---

> ### Author Response · Authors · 2025-07-31
> **Response to Reviewer Feedback**
>
> Thank you for your response! We appreciate your adept summary of our work, CGBench, and the strengths you mentioned. We agree that this work is impactful and opens up avenues for future work in LLM reasoning for clinical genetics and other domain-specific applications. In the following, we will address each of your comments from the “Limitations Weaknesses” (LW) section and the “Additional Feedback” (AF) section, broken down by the order of comments.
>
> LW1 ("$C_{vcep}$ and $\hat{e}^y_k$..."): Thank you for pointing this out, we have fixed this in our revised manuscript as the previous description had incorrect notation. This section (portion of Section 3.3) will be updated in the revised manuscript.
>
> LW2 ("In Table 1..."): Thank you for pointing out this initially counterintuitive result. Specifically this refers to the result for Primary code prediction for the Random model. The random baseline here is one that uniformly selects from the options given for this level of codes. At the “Primary” level, as seen in Figure 2, the model simply chooses from “Pathogenic” or “Benign”, a binary option. We then sample 5 times to reflect the sampling done for the language models. Recall intuitively measures “out of all of your samples, did you sample the correct code”, so the probability of “failure” in this case amounts to sampling the same incorrect code 5 times in Bernoulli trials with p=0.5, which is $1 - 0.5^5 = 0.96875$. The language models have lower performance because they often sample along one primary code as their sampling is not uniform.
>
> LW3 ("The two VCI..."): Thank you for commenting on this important point. Precisely, the total sample sizes for the tasks we present in the VCI are 239 for evidence scoring and 286 for evidence verification. This number is comparable to similar human-written (e.g., not automatically mined or generated) benchmarks of scientific capabilities of language models. LitQA2 [1], a question-answering benchmark on scientific papers, is 248 samples and GPQA [2], a standard benchmark in challenging scientific tasks, is 448 samples, with GPQA-Diamond, the most challenging set, being 198 samples. This puts CGBench's VCI tasks as comparably-sized and the GCI task as much larger, at 2,155 samples.
>
> AF1 ("While the abstract..."): Thank you for this suggestion, we’ve shortened our abstract in our revised manuscript to be more succinct. Our new abstract reads as follows:
> "Variant and gene interpretation are fundamental to personalized medicine and translational biomedicine. However, traditional approaches are manual and labor-intensive. Generative language models (LMs) can facilitate this process, accelerating the translation of fundamental research into clinically-actionable insights. While existing benchmarks have attempted to quantify the capabilities of LMs for interpreting scientific data, these studies focus on narrow tasks that do not translate to real-world research. To meet these challenges, we introduce CGBench, a robust benchmark that tests reasoning capabilities of LMs on scientific publications. CGBench is built from ClinGen, a resource of expert-curated literature interpretations in clinical genetics. CGBench measures the ability to 1) extract relevant experimental results following precise protocols and guidelines, 2) judge the strength of evidence, and 3) categorize and describe the relevant outcome of experiments. We test 8 different LMs and find that while models show promise, substantial gaps exist in literature interpretation, especially on fine-grained instructions. Reasoning models excel in fine-grained tasks but non-reasoning models are better at high-level interpretations. Finally, we measure LM explanations against human explanations with an LM judge approach, revealing that models often hallucinate or misinterpret results even when correctly classifying evidence. CGBench reveals strengths and weaknesses of LMs for precise interpretation of scientific publications, opening avenues for future research in AI for clinical genetics and science more broadly."
> This reduces the original abstract at 289 words to 217 words, a 25\% reduction in length.
>
> References:
> [1] Skarlinski et al., "Language agents achieve superhuman synthesis of scientific knowledge", arXiv 2024
> [2] Rein et al., "GPQA: a graduate-level Google-proof Q\&A benchmark", arXiv 2023

---

> > ### Comment · Reviewer_GmP3 · 2025-08-06
> >
> > Thank you to the authors for the rebuttal, which has addressed my questions. I remain positive about the paper and will keep my score.

---

> > > ### Author Response · Authors · 2025-08-06
> > >
> > > Thank you again for your valuable feedback!

---

### Comment · Area_Chair_vypg · 2025-08-04
**Please check the authors response**

Dear reviewers,

Thanks for reviewing this paper. Could you check if the author's response has addressed your concerns? Feel free to raise any further questions if you have. Please note that acknowledgement is mandatory if you haven't done so already.

Best,

AC

---

### Author Response · Authors · 2025-08-06
**Thank you to reviewers - request for additional feedback**

Thank you to all the reviewers for your insightful feedback on our work! We appreciate that reviewers praise our work for being “built on high-quality resources” (Rev. 4ngH) with tasks that “follow real-world scientific protocols used in clinical genetics” (Rev. GmP3). Reviewers also highlight our central contributions, noting that CGBench “matches actual curation workflow” (Rev. rfNg) and **“pushes the evaluation beyond simple information retrieval to assess deeper reasoning”** (Rev. nJj5). Finally, reviewers note that the tasks in CGBench provide “interesting direction for future discussion” (Rev. GmP3) and “a good starting point for progressing this research direction” (Rev. 4ngH).

In addition to praises, we also thank the reviewers for their nuanced and detailed critiques of our work. We have prepared detailed responses to each of your reviews that can be found below each of your reviews below. We’d like to highlight some of our major responses as a result of your suggestions:

1. In response to Reviewer nJj5’s comment on the inconsistencies of the in-context learning examples, we have performed a robustness analysis on the selection of in-context examples, which shows a **high level of stability in results** when changing the examples used in the prompts as demonstrations to the LLMs.
2. Based on Reviewer rfNg’s comment on if CGBench examples might have been seen by the proprietary LLMs used in this work, we have conducted experiments using PubMed IDs as a control for the queries in the VCI evidence scoring task. Our results show that by removing the full-text of the papers and using instead PubMed IDs, performance on tertiary code prediction drops by over 50%. This is important given the construction of the webpages from which CGBench was sourced, which we discuss in detail in the response.
3. In addition, based on Reviewer rfNg’s suggestion, we have performed analysis on examples from before and after the cutoff dates for our LLMs used in the study. Our results show a definite drop in performance for tertiary code prediction. **Such an analysis shows the flexibility of CGBench in allowing for analyses that stratify by date of annotation.**
4. Finally, we have provided further clarifications to each of the reviewers on many points within our manuscript, including the relative size of CGBench compared to other common literature interpretation benchmarks (Rev. GmP3), the similarity of codes across VCEPs (Rev. rfNg), the role of LM judge’s in the case of CGBench as a way to measure correspondence with human annotators (Rev. nJj5), and the scope of CGBench on literature interpretation (Rev. 4ngH).

We invite any other comments, especially from reviewers who have not yet had the chance to respond, on our work or requests for clarification on our benchmarking, results, or any other details in the paper. Thank you again for your time and effort to help us improve CGBench.

---

### Note · Authors · 2025-08-12

Reviewers and ACs,

Thank you for your comments on our work! We appreciate the feedback on our paper, and we have responded with detailed rebuttals to each review below. We have summarized our major responses to the review in our most recent global "official comment", including specific experiments that we have ran for reviewers' comments. In this same comment, we summarize the overall positive reviews of our work. Overall, reviewers gave very positive marks on our paper while identifying a few additional steps in communication and analysis that we were able to add to our work.

In response to reviewer feedback, we will make several minor changes to the work that we refer to in our rebuttals, including small changes to notation, shortening of the abstract, further clarification on the similarities of VCEPs, robustness results for ICL experiments, and details on data contamination experiments. We believe these changes will further boost the communication and full evaluation of CGBench.

In conclusion, we believe CGBench is a robust, challenging benchmark to understand interpretation of scientific literature by LMs. Our unique source of data, gathered from ClinGen, provides detailed, precise data gathered from human experts, and we leverage this to evaluate LMs in a real-world scientific setting. Our work reveals multiple directions for future research, including in improving the discriminative capabilities of LMs for understanding the quality of scientific evidence and aligning the interpretation of scientific evidence closer to that of researchers. This benchmark is of great importance to the AI community in particular, providing an openly-available, easy-to-use resource of high-quality human annotations on scientific literature that can be easily integrated into modern LM infrastructures.

Thank you,

Authors

---

### Decision · Program_Chairs · 2025-09-18

**Decision:**

Accept (poster)

**Comment:**

This paper presents CGBench, a new benchmark designed to test how well language models can reason about scientific information in the field of clinical genetics. The benchmark is built using expert-reviewed literature and includes tasks that reflect real challenges faced by professionals in this domain. Reviewers found the work to be well-motivated and relevant, especially for evaluating model performance in high-stakes areas like healthcare. The paper also includes a detailed analysis of model errors and hallucinations, which adds to its practical value.

During the discussion phase, the authors responded clearly and thoroughly to reviewer concerns, including questions about how generalizable the benchmark is and how the tasks are defined. Reviewers appreciated the clarifications and maintained positive ratings. Overall, the paper makes a meaningful contribution to both the AI and biomedical communities.